# Dopamine neurons projecting to the posterior striatum form an anatomically distinct subclass

William Menegas[1], Joseph F Bergan[1†], Sachie K Ogawa[1‡], Yoh Isogai[1], Kannan Umadevi Venkataraju[2], Pavel Osten[2], Naoshige Uchida[1], Mitsuko Watabe-Uchida[1*]

[1]Center for Brain Science, Department of Molecular and Cellular Biology, Harvard University, Cambridge, United States; [2]Cold Spring Harbor Laboratory, Cold Spring Harbor, United States

*For correspondence: mitsuko@mcb.harvard.edu

Present address: †Department of Psychological and Brain Sciences, University of Massachusetts Amherst, Amherst, United States; ‡RIKEN-MIT Center for Neural Circuit Genetics at the Picower Institute for Learning and Memory, Department of Biology and Department of Brain and Cognitive Sciences, Massachusetts Institute of Technology, Cambridge, United States

**Abstract** Combining rabies-virus tracing, optical clearing (CLARITY), and whole-brain light-sheet imaging, we mapped the monosynaptic inputs to midbrain dopamine neurons projecting to different targets (different parts of the striatum, cortex, amygdala, etc) in mice. We found that most populations of dopamine neurons receive a similar set of inputs rather than forming strong reciprocal connections with their target areas. A common feature among most populations of dopamine neurons was the existence of dense 'clusters' of inputs within the ventral striatum. However, we found that dopamine neurons projecting to the posterior striatum were outliers, receiving relatively few inputs from the ventral striatum and instead receiving more inputs from the globus pallidus, subthalamic nucleus, and zona incerta. These results lay a foundation for understanding the input/output structure of the midbrain dopamine circuit and demonstrate that dopamine neurons projecting to the posterior striatum constitute a unique class of dopamine neurons regulated by different inputs.

## Introduction

A longstanding challenge in neuroscience is to understand how neurons compute their output by integrating information from multiple sources. Mapping precise neural connectivity is a critical step towards understating neural computation. Recent developments in viral, molecular, and imaging techniques offer unprecedented opportunities to outline various aspects of the brain's wiring diagram in systematic and quantitative manners (*Denk et al., 2012*; *Watabe-Uchida et al., 2012*; *Huang and Zeng, 2013*; *Osten and Margrie, 2013*; *Pollak Dorocic et al., 2014*; *Hart et al., 2014*; *Ogawa et al., 2014*; *Oh et al., 2014*; *Weissbourd et al., 2014*; *Callaway and Luo, 2015*).

Midbrain dopamine neurons play important roles in various brain functions including motivation, reinforcement learning, and motor control (*Wise, 2004*; *Redgrave and Gurney, 2006*; *Ikemoto, 2007*; *Schultz, 2007*). Recording experiments have shown that many dopamine neurons signal reward prediction error (RPE): the discrepancy between predicted and actual reward value (*Schultz et al., 1997*; *Bayer and Glimcher, 2005*; *Bromberg-Martin et al., 2010*; *Clark et al., 2012*; *Cohen et al., 2012*; *Schultz, 2015*). Classically, it is thought that RPE coding is relatively uniform among dopamine neurons, and that dopamine's major function is to guide behavior toward maximizing future rewards. However, recent studies have suggested that there are at least two types of dopamine neurons, value-coding and salience-coding (*Matsumoto and Hikosaka, 2009*), although the extent of physiological diversity remains controversial (*Fiorillo, 2013*; *Fiorillo et al., 2013a*, *2013b*).

**eLife digest** Most neurons send their messages to recipient neurons by releasing a substance called a 'neurotransmitter' that binds to receptors on the target cell. The sites of this type of signal transmission are called synapses. Some small populations of neurons modulate the activity of hundreds or thousands of these synapses all across the brain by releasing 'neuromodulators' that affect how they work. These neuromodulators are essential because they broadcast information that is likely to be useful to many brain regions, like a 'news channel' for the brain.

One important neuromodulator in the mammalian brain is dopamine, which contributes to motivation, learning, and the control of movement. Clusters of cells deep in the brain release dopamine, and people with Parkinson's disease gradually lose these cells. This makes it increasingly difficult for their brains to produce the correct amount of dopamine, and results in symptoms such as tremors and stiff muscles.

Individual dopamine neurons typically send information to a single part of the brain. This suggests that dopamine neurons with different targets might have different roles. To explore this possibility, Menegas et al. classified dopamine neurons in the mouse brain into eight types based on the areas to which they project, and then mapped which neurons send input signals to each type. These inputs are likely to shape the activity of each type (that is, their 'message' to the rest of the brain). The mapping revealed that most dopamine neurons do not receive substantial input from the area to which they project (i.e., they do not form 'closed loops'). Instead, most of their input comes from a common set of brain regions, including a particularly large number of inputs from the ventral striatum.

However, Menegas et al. found one exception. Dopamine neurons that target part of the brain called the posterior striatum receive relatively little input from the ventral striatum. Their input comes instead from a set of other brain structures, and in particular from a region called the subthalamic nucleus. Electrical stimulation of the subthalamic nucleus can help to relieve the symptoms of Parkinson's disease. Therefore, the results presented by Menegas et al. suggest that this population of dopamine neurons might be particularly relevant to Parkinson's disease and that focusing future studies on them could ultimately be beneficial for patients.

Most dopamine neurons reside in midbrain nuclei called the ventral tegmental area (VTA), the substantia nigra pars compacta (SNc), and the retrorubral field (RRF). Clusters of dopamine neurons in and around these nuclei are designated A10, A9, and A8, respectively. In the mouse brain, just ~30,000 dopamine neurons reside in these nuclei (*Zaborszky and Vadasz, 2001*). As with other monoamine neurotransmitters in the brain such as serotonin and noradrenaline, this small population of midbrain dopamine neurons exerts its influence over much of the brain as a neuromodulator. However, compared to the other monoamine neuromodulators, the dopamine system is anatomically unique in that the collateralization of dopamine neurons is limited (*Moore and Bloom, 1978*; *Swanson, 1982*; *Sobel and Corbett, 1984*). In other words, unlike the other monoaminergic neurons, a single dopamine neuron tends to target just one brain area, raising the possibility that different functional populations can be defined by the region that they target.

Recent studies have shown that dopamine neurons projecting to different targets have distinct properties (*Lammel et al., 2008*, *2011*, *2012*, *2014*; *Kim et al., 2014*). For instance, a population of dopamine neurons in the posterior medial VTA that projects to the medial prefrontal cortex (mPFC) in mice, is unique with respect to many features: low expression of dopamine transporter (DAT), tyrosine hydroxylase (TH), and the D2 dopamine receptor, narrow spike waveform, high baseline firing and low level of cocaine-induced synaptic plasticity (*Lammel et al., 2008*, *2011*, *2012*, *2014*). Dopamine neurons that project to different areas are located at slightly different, but highly overlapping locations in the midbrain (*Swanson, 1982*; *Yetnikoff et al., 2014*). These results have raised the possibility that projection target, rather than anatomical location, can better classify functional groups of dopamine neurons.

To understand how dopamine neurons compute their output, it is critical to know their inputs. Early systematic studies produced a thorough list of input areas to dopaminergic nuclei (*Geisler and Zahm, 2005*; *Geisler et al., 2007*) using conventional tracers. A recent study using a cell-type specific

transsynaptic tracing method employing a rabies virus confirmed that many of these areas actually project directly onto dopamine neurons (*Watabe-Uchida et al., 2012*). This study also showed that VTA dopamine neurons received a distinct set of inputs compared to SNc dopamine neurons. It is possible that the differences observed in this study between inputs to VTA and SNc dopamine neurons were attributable to the uneven distribution of populations of dopamine neurons with different projection targets between these two nuclei. Both VTA and SNc contain dopamine neurons projecting to different targets, and each population of dopamine neurons projecting to a given target is distributed in a complex manner, often encompassing both VTA and SNc. Thus, it remains to be determined whether dopamine neurons projecting to different targets receive different or similar inputs.

In the present study, we classified dopamine neurons according to their projection targets and defined the distribution of monosynaptic inputs of each subpopulation. We compared the inputs of dopamine neurons defined by eight major targets: different parts of the striatum (ventral striatum [VS], dorsal striatum [DS], tail of the striatum [TS]), cortex (mPFC, orbitofrontal cortex [OFC]), central amygdala (Amy), globus pallidus (GP), and lateral habenula (lHb). To collect and analyze this large data set, we developed a data acquisition and analysis pipeline using a brain clearing method (CLARITY) (*Chung and Deisseroth, 2013*), whole-brain imaging using light-sheet microscopy (*Keller et al., 2010*), and semi-automated software for analysis. We found that populations of dopamine neurons projecting to most of these targets receive a similar set of inputs, while dopamine neurons projecting to TS ('tail of the striatum' or 'posterior striatum') are a clear outlier. These results indicate that classifying dopamine neurons based on their projection target is a viable approach, and lay a foundation for studying the mechanisms by which dopamine neurons are regulated. Our approach is based on an automated imaging and analysis suite that will facilitate similar anatomical studies in other brain systems in an efficient and quantitative manner.

## Results

### Tracing monosynaptic inputs to projection-specific dopamine neurons

To visualize the monosynaptic inputs to dopamine neurons, we used a retrograde transsynaptic tracing system based on a modified rabies virus (SADΔG-EGFP(EnvA)) (*Wickersham et al., 2007*). This rabies virus is pseudotyped with an avian sarcoma and leukemia virus (ALSV-A) envelope protein (EnvA), so that the initial infection is restricted to cells that express a cognate receptor (TVA) in mammalian brains. In addition, this rabies virus lacks the gene encoding the rabies virus envelope glycoprotein (RG), which is required for trans-synaptic spread. This allows us to restrict trans-synaptic spread to cells that exogenously express RG. In this way, only monosynaptic (but not poly-synaptic) inputs are trans-synaptically labeled. In our previous work, we used two helper viruses to express TVA and RG under the control of Cre recombinase (AAV5-FLEX-TVA-mCherry and AAV8-FLEX-RG) (*Watabe-Uchida et al., 2012*). We previously injected these helper viruses into the VTA or SNc of transgenic mice expressing Cre specifically in dopamine neurons (dopamine transporter-Cre, or DAT-Cre) (*Backman et al., 2006*) to specifically label the monosynaptic inputs of these dopamine neurons throughout the brain (*Watabe-Uchida et al., 2012*).

In the present study, we sought to restrict the initial rabies infection to subpopulations of dopamine neurons defined by their projection sites. To do this, we injected a helper virus that expresses TVA under the control of Cre recombinase (AAV5-FLEX-TVA) into a dopamine projection site and co-injected a virus bearing blue fluorescent protein (AAV1-CA-BFP) to mark this site. Because it has been shown that minute TVA expression is sufficient for infection by ALSV-A (*Belanger et al., 1995*), we reasoned that DAT-Cre-expressing dopamine neurons in the midbrain which were retrogradely infected by AAV5-FLEX-TVA would express enough TVA for initial rabies infection in a Cre-dependent manner (*Figure 1A*). To verify this, we injected AAV5-FLEX-TVA into one of the projection sites, the DS. After 3 weeks, rabies virus was injected into both VTA and SNc. In this experiment, we did not inject AAV8-FLEX-RG, so the rabies virus did not spread trans-synaptically. We did this to visualize only neurons that were directly infected by rabies virus (*Figure 1B*). Consistent with our intention, we observed that many VTA/SNc neurons were infected by the rabies virus and therefore expressed GFP (*Figure 1B*; *Figure 1—figure supplement 1A*). Based on antibody staining, almost all of the GFP-positive neurons were also positive for TH (*Figure 1C–F*), a marker for dopamine neurons (98 ± 1%; $n = 600$ neurons, $n = 3$ mice). Given that injecting pseudotyped rabies virus alone

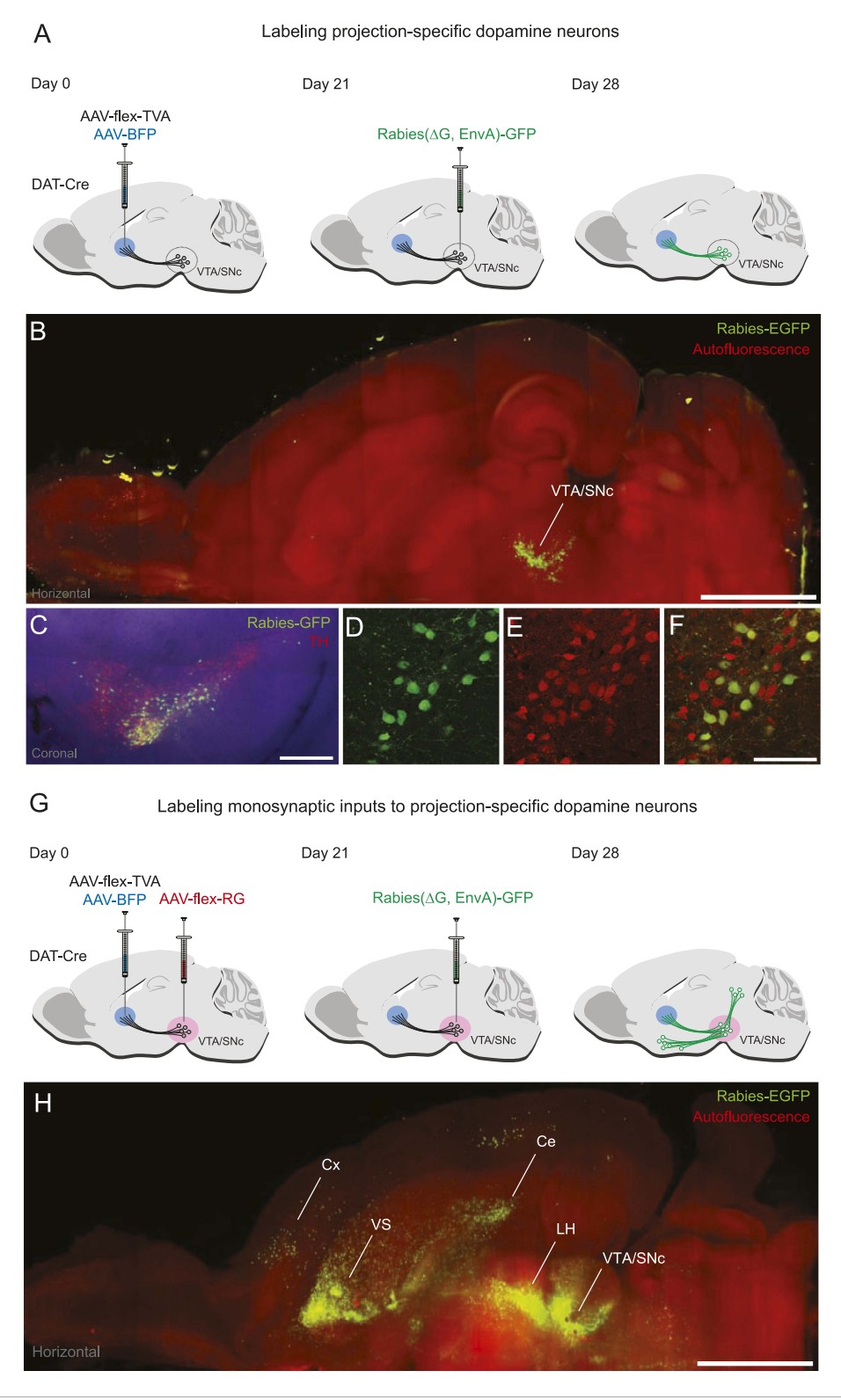

**Figure 1**. Labeling projection-specific dopamine neurons and their monosynaptic inputs throughout the brain with rabies-GFP. (**A**) A schematic of the injections used to label projection-specific populations of dopamine neurons. The blue circle represents the site of infection with adeno-associated virus (AAV)-FLEX-TVA and green neurons represent the DAT-Cre-expressing dopamine neurons projecting to that area. (**B**) Horizontal optical section showing
*Figure 1. continued on next page*

*Figure 1. Continued*

rabies-GFP signal following AAV-FLEX-TVA injection into the striatum of a DAT-Cre animal followed by rabies injection into the ventral tegmental area (VTA) and substantia nigra pars compacta (SNc). The numbers of infected neurons are shown in *Figure 1—figure supplement 1*. Bar indicates 2 mm. (**C**) Coronal physical section showing rabies labeled neurons from (**B**) in green and anti-TH antibody staining in red. Bar indicates 500 µm. (**D**–**F**) Higher magnification image of rabies labeled neurons and tyrosine hydroxylase (TH) staining. Bars indicate 200 µm. (**G**) A schematic of the injections used to label the inputs of projection-specific populations of dopamine neurons throughout the brain. The blue circle represents the site of infection with AAV-FLEX-TVA and the red circle represents the site of infection with AAV-FLEX-RG. Green neurons outside of VTA/SNc represent the monosynaptic inputs labeled throughout the brain. (**H**) Horizontal optical section showing rabies-GFP signal following AAV-FLEX-TVA injection into the striatum and AAV-FLEX-RG into the VTA and SNc of a DAT-Cre animal followed by rabies injection into the VTA and SNc. Number of infected neurons shown in *Figure 1—figure supplement 1*. Bar indicates 2 mm.

The following figure supplement is available for figure 1:

**Figure supplement 1**. Number of starter cells and inputs labeled.

---

resulted in very few infections (*Watabe-Uchida et al., 2012*), these results indicate that the injection of AAV5-FLEX-TVA into a dopamine projection site allowed us to restrict infection of rabies virus to dopamine neurons in a projection specific manner.

To allow rabies virus to spread trans-synaptically, we performed the same AAV5-FLEX-TVA injection to infect DS-projecting dopamine neurons, but this time also injected AAV8-FLEX-RG into both the VTA and SNc. Because a large quantity of RG is required for robust trans-synaptic spread, we injected this virus near the cell bodies of dopamine neurons (in the VTA and SNc) rather than near their axons. After 3 weeks, rabies virus was injected into both the VTA and SNc (*Figure 1G*). This resulted in a large number of trans-synaptically labeled GFP-positive neurons outside VTA/SNc (*Figure 1H*; *Figure 1—figure supplement 1B*). Together, these results demonstrate that our method allows us to label subpopulations of dopamine neurons defined by their projection sites and the monosynaptic inputs to these populations. We used this method to identify the monosynaptic inputs to dopamine neurons projecting to different targets.

## Acquisition and analysis of rabies tracing data

Comparing the eight experimental groups required us to acquire and process a large data set. To achieve this goal, we developed a new data acquisition and analysis suite. In order to quantify the spread of rabies virus in each condition, we cleared brains using CLARITY to prepare them for light sheet microscopy (*Figure 2A*). Before imaging each brain, we pre-screened to ensure sufficient optical clarity by shining a 488 nm light sheet through the ventral part of the brain and collecting light through an objective near the dorsal surface (∼6 mm away). 77 of 89 processed brains (∼87%) were sufficiently clear for visualization of the ventral-most cells, and only these brains were used. We imaged each brain from the dorsal and ventral sides (such that each image was a 'horizontal' optical section), and then merged these images to create a continuous 3D image of the brain (*Figure 2B*).

Because rabies spreads to monosynaptic input neurons, the only information we ultimately needed to collect from each brain for the present goal of comparing the distribution of inputs was the position of each labeled cell body. Therefore, we only needed to image with sufficient resolution to distinguish every cell body throughout the brain (*Figure 2C*). This resulted in ∼2 Tb of image data from each brain, allowing us to image many brains, whereas imaging with even 2× higher magnification in each dimension would have led to prohibitively (based on our present processing capacity) large file sizes of ∼20 Tb per brain. We applied multiple segmentation algorithms to this data to classify each pixel as 'cell' or 'non-cell' (*Figure 2D*; see 'Materials and methods'). Quality of segmentation was consistent across many regions and imaging depths, including cortex (*Figure 2E*), striatum (*Figure 2F*), midbrain (*Figure 2G*), and cerebellum (*Figure 2H*). Because each cell was present in several images (at different depths), we then collapsed all neighboring 'cell' pixels in all three dimensions to generate a list of the centroids of the detected cells. To compare the distribution of cells across brains, we aligned the autofluorescence image captured from each brain to a common 'reference space' and used these

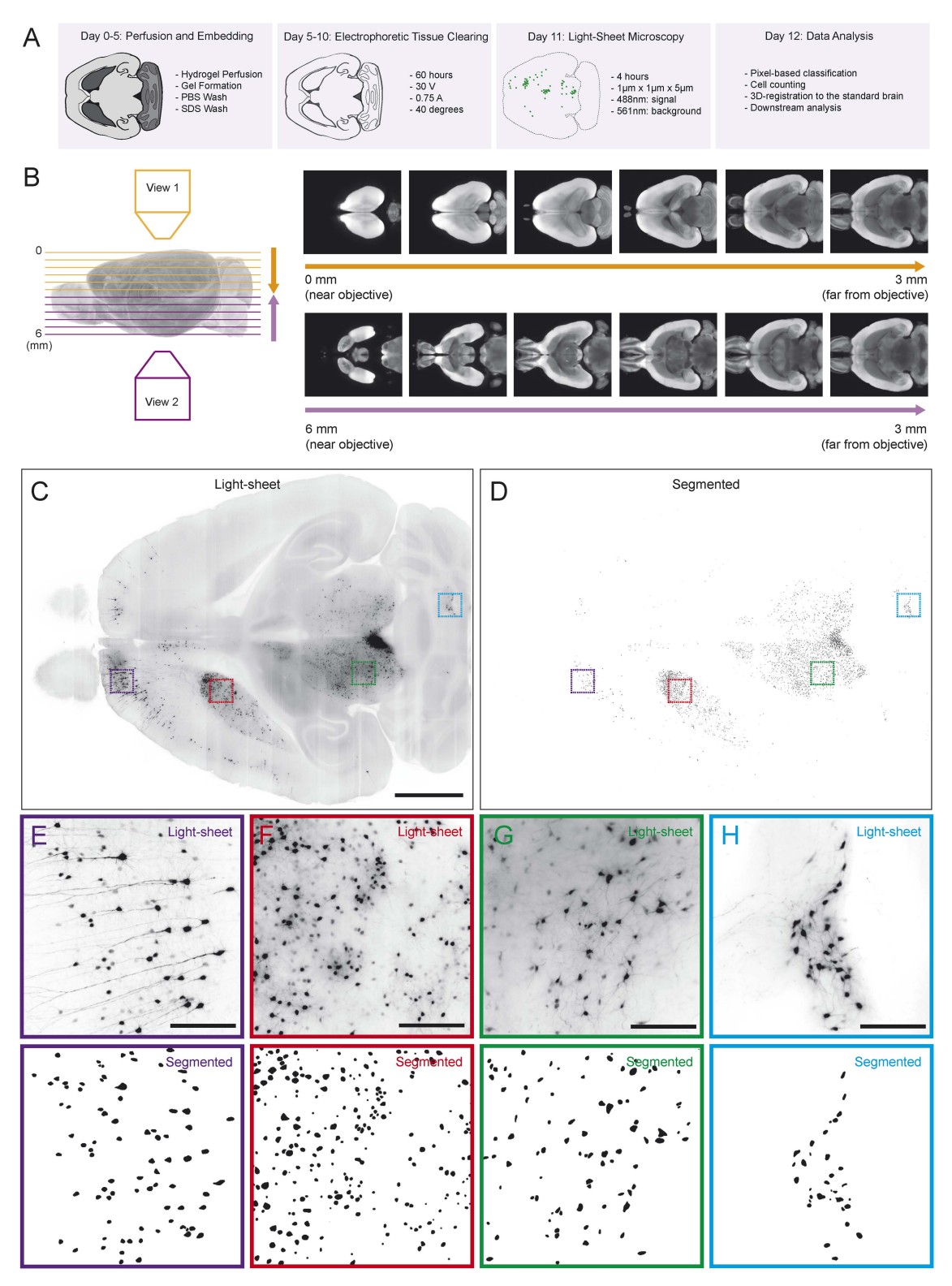

**Figure 2**. Automated acquisition and analysis of whole-brain tracing data. (**A**) A schematic of the brain clearing, imaging, and analysis pipeline used to acquire data from brains labeled using the injection schematics outlined in *Figure 1A* and *Figure 1G*. (**B**) A graphical explanation of the image acquisition and stitching process. Whole brains were imaged horizontally, from the dorsal (orange arrow) and ventral (purple arrow) sides and these images were stitched and combined to create a whole brain image. An example brain expressing tdTomato under the control of genetically encoded Vglut2-Cre is

*Figure 2. continued on next page*

*Figure 2. Continued*

shown. (**C**) An example of a horizontal optical section acquired as described in **A**, **B**. Boxes indicate the locations of inset panels. Bar indicates 2 mm. (**D**) The automatically generated segmentation of **C**, with each labeled cell being represented by a single pixel. Bar indicates 2 mm. (**E**–**H**) Insets displaying raw images and automatically generated segmentations from the indicated regions (cortex, striatum, midbrain, and cerebellum). Bars indicate 200 μm.

transformations to align each set of centroids into this common space (see 'Materials and methods'). By aligning brains with different genetically defined populations of neurons labeled, we were able to determine the boundaries of major brain regions in this reference space in three dimensions. This strategy for analysis mirrors the analysis pipeline called 'CUBIC', proposed for comparable analyses of whole-brain images acquired using a similar clearing method (*Susaki et al., 2014*).

## Distribution of dopamine neurons projecting to different projection targets

The main target structure of midbrain dopamine neurons is the striatum. However, dopamine neurons also project to other brain areas including other basal ganglia structures, amygdala, and much of the cortex. We characterized the distribution of dopamine neurons that project to cortex (mPFC, OFC), striatum (VS, DS, TS), GP, central amygdala (Amy) and lateral habenula (lHb). To do this, AAV5-FLEX-TVA was injected into one of the eight projection sites in DAT-Cre animals expressing Cre specifically in dopamine neurons (without injecting AAV8-FLEX-RG into the midbrain). 3 weeks later, rabies virus was injected into both VTA and SNc to infect dopamine neurons projecting to the area of the AAV5-FLEX-TVA injection (*Figure 1A*). Brain samples were collected after 1 week.

As previously reported, we found that dopamine neurons with distinct projection targets reside in different, but overlapping, areas of the midbrain (*Figure 3*; *Figure 3—figure supplement 1*; *Figure 3—figure supplement 2*; *Figure 3—figure supplement 3*) (*Swanson, 1982*; *Bjorklund and Dunnett, 2007*; *Lammel et al., 2008*; *Haber, 2014*). Interestingly, we observed an overlapping but dorso-laterally shifted distribution of dopamine neurons that project to VS, DS, and TS, in this order (*Figure 3—figure supplement 1*; *Figure 3—figure supplement 2*). Because many of the labeled populations of neurons overlapped in space, it is difficult to discriminate between populations based on location alone. In the case of Amy-projecting dopamine neurons, we observed labeling throughout VTA/SNc and also substantial labeling in supramammillary areas (A10rv) (*Yetnikoff et al., 2014*). In short, each brain area receives projections from dopamine neurons in slightly different, but highly overlapping subareas of VTA/SNc. Furthermore, we found that the distribution of labeled neurons in the VTA/SNc was similar in cases with and without transsynaptic spread (with and without RG, respectively), indicating that transsynaptic spread between primarily infected dopamine neurons and other dopamine neurons did not lead to the nonspecific infection of all dopamine neurons (*Figure 3—figure supplement 3*).

## Distribution of inputs: TS-projecting dopamine neurons receive a unique set of inputs compared to other populations

Next, we mapped the monosynaptic inputs to these eight subpopulations of dopamine neurons (*Figure 3—figure supplement 4*; *Figure 3—figure supplement 5*) defined by their projection target. Because the number of labeled neurons is different in each condition (*Figure 1—figure supplement 1*), comparing different conditions required a method of normalization. We first observed the distributions of labeled neurons by plotting the positions of labeled neurons for each condition. To normalize for the different numbers of total neurons labeled for visualization, we randomly sampled 1500 neurons from each brain, and plotted corresponding neurons in coronal sections (*Figure 3*).

In our previous studies, we mapped the monosynaptic inputs to all of the dopamine neurons in VTA and SNc (*Watabe-Uchida et al., 2012*; *Ogawa et al., 2014*). We initially hypothesized that each of the eight populations would receive inputs from a subset of the regions identified in these previous studies. Surprisingly, we found that the distribution of inputs to each population appeared to be largely similar (*Figure 3*). We found that monosynaptic inputs to most dopamine neurons were concentrated in the ventral striatum (*Figure 3A*) and that several areas containing inputs are continuous around the medial forebrain bundle, along the entire anterior–posterior axis (across the

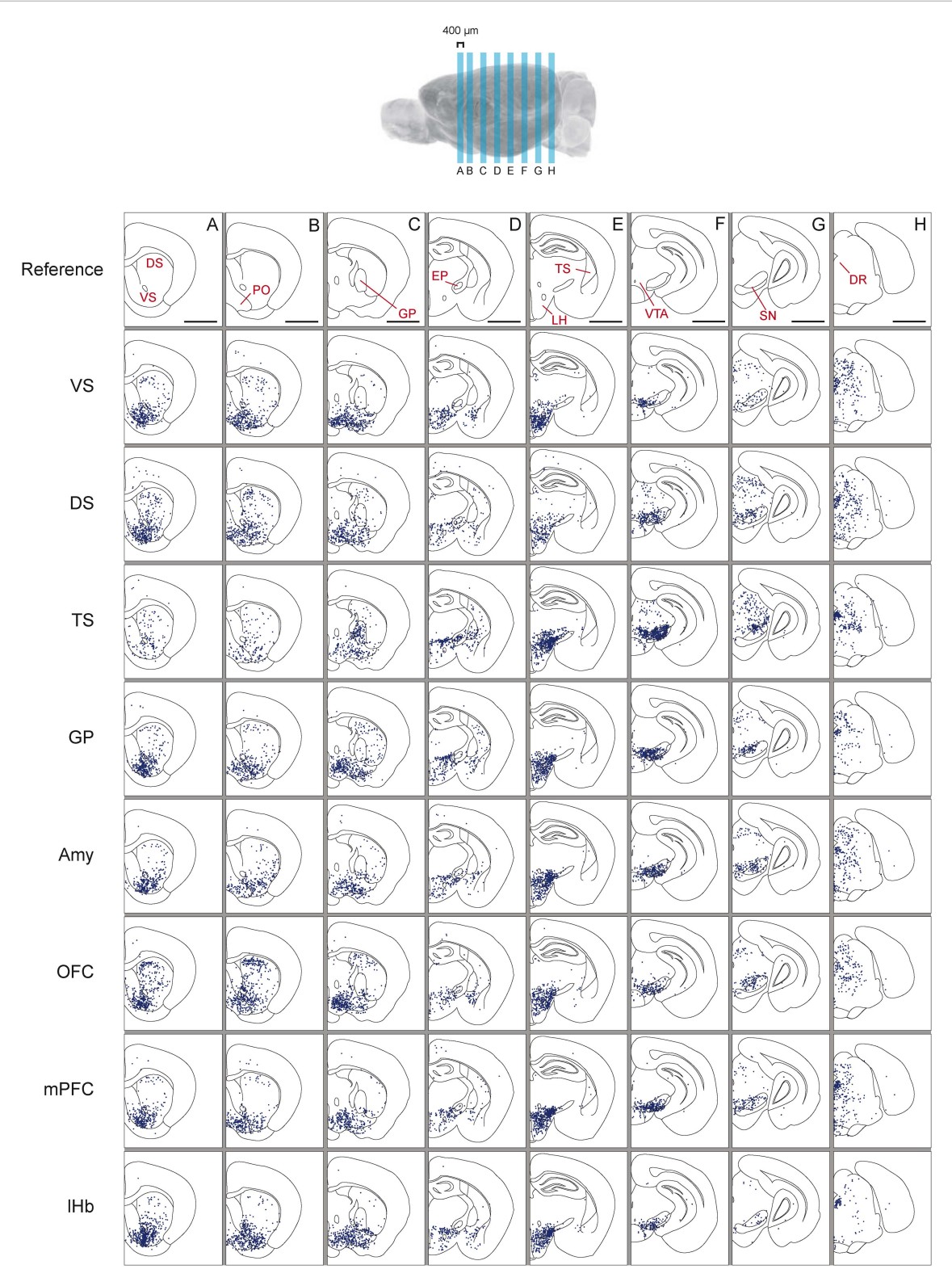

**Figure 3**. Distribution of monosynaptic inputs to projection-specific populations of dopamine neurons throughout the brain. A summary of the inputs to each of the projection-specified populations of dopamine neurons assayed (using the injection scheme from *Figure 1G*) normalized such that the 1500 neurons were randomly sampled from each brain. We then combined such subsampled neurons from three randomly selected animals for each condition and plotted them in corresponding 400 µm sections. Inputs to: VS-projecting, DS-projecting, TS-projecting, GP-projecting, Amy-projecting,
*Figure 3. continued on next page*

*Figure 3. Continued*

OFC-projecting, mPFC-projecting, and lHb-projecting dopamine neurons as well as selected coronal sections for reference (**A–H**). In this figure, and all others, the following abbreviations were used: 'VS' for ventral striatum, 'DS' for anterior dorsal striatum, 'TS' for tail of the striatum (posterior striatum), 'GP' for globus pallidus, 'Amy' for amygdala, 'OFC' for orbitofrontal cortex, 'mPFC' for medial prefrontal cortex, and 'lHb' for lateral habenula. Data collected in the same manner from experiments with no transsynaptic spread shown in *Figure 3—figure supplement 1* and *Figure 3—figure supplement 2*, comparison of the two conditions shown in *Figure 3—figure supplement 3*, and injection sites used shown in *Figure 3—figure supplement 4* and *Figure 3—figure supplement 5*. Bars indicate 2 mm.

The following figure supplements are available for figure 3:

**Figure supplement 1**. Distribution of projection-specific populations of dopamine neurons within the midbrain.

**Figure supplement 2**. Distribution of projection-specific populations of dopamine neurons within the midbrain: Maximum Intensity Projection.

**Figure supplement 3**. Comparison of labeled cell distribution between control and experimental groups.

**Figure Supplement 4**. Injection Sites.

**Figure supplement 5**. Subdivisions of the striatum.

nucleus boundaries) (*Figure 3A–E*) as was observed previously (*Geisler and Zahm, 2005*; *Watabe-Uchida et al., 2012*), regardless of the projection targets. However, we found that TS-projecting dopamine neurons did not show a similar concentration of inputs: these neurons received many fewer inputs from anterior regions (*Figure 3A,B*) and ventromedial regions (*Figure 3C,D*). In short, these observations suggested that the overall distribution of inputs to TS-projecting dopamine neurons is different from those of the rest of the dopamine neuron populations (whose inputs largely overlapped with one another). We next examined these similarities and differences in more systematic and quantitative manners.

To quantify the differences between inputs to dopamine neurons with different projection sites, we compared the percentage of input neurons observed in each brain area in each condition (*Figure 4*). This normalization allowed us to average between brains within a given condition ($n$ = 3–5 mice per condition, $n$ = 4 for TS) and then to compare across conditions. Although we found some differences between subpopulations, the overall patterns were surprisingly similar. Among the brain systems, we found that the basal ganglia and hypothalamus provided the majority of inputs to dopamine neurons (*Figure 4*, *Figure 4—figure supplement 1*). In the basal ganglia, the ventral striatum (nucleus accumbens core and shell), DS and ventral pallidum were the largest sources of inputs. In hypothalamus, the lateral hypothalamus (LH) (including the parasubthalamic nucleus [PSTh]) contained the largest number of inputs, and the preoptic areas also provided a substantial number. In the midbrain, the superior colliculus and dorsal raphe provided inputs as well.

In spite of similarities among dopamine populations, several prominent differences emerge upon focusing on TS-projecting dopamine neurons. While the ventral striatum is one of the largest sources of inputs to most of the subpopulations, it is a relatively minor source of inputs to TS-projecting dopamine neurons (*Figure 3A*, *Figure 4*). Instead, TS-projecting neurons receive an increased proportion of their inputs from the GP (*Figure 3C*, *Figure 4*), entopeduncular nucleus (EP, *Figure 3D*, *Figure 4*), PSTh, subthalamic nucleus (STh) and zona incerta (ZI) (*Figure 3E*, *Figure 4*). To statistically verify these observations, we first performed a 1-way analysis of variance (ANOVA) among the 20 main input areas (*Figure 4—figure supplement 1*). After correcting for multiple comparisons, nucleus accumbens shell, GP, STh and ZI remained significant ($p$ = 0.00045, 0.0017, 0.00023, and 0.0028, respectively; significant after Holm-Sidak corrections for multiple comparisons). These differences are mainly attributable to the effect of the TS-projecting dopamine population. If we remove TS in the analysis, none of these areas retain statistical significance with the exception of the ventromedial hypothalamus ($p$ = 0.0028, 1-way ANOVA), which appears to be a slightly larger source of inputs to VS-projecting dopamine neurons. Furthermore, in all of the above cases, the percentage of inputs from TS was significantly larger than the group mean of the remainder of the populations ($p$ < 0.001; t-test). These results suggest that many dopamine neurons are embedded in a 'canonical' circuit

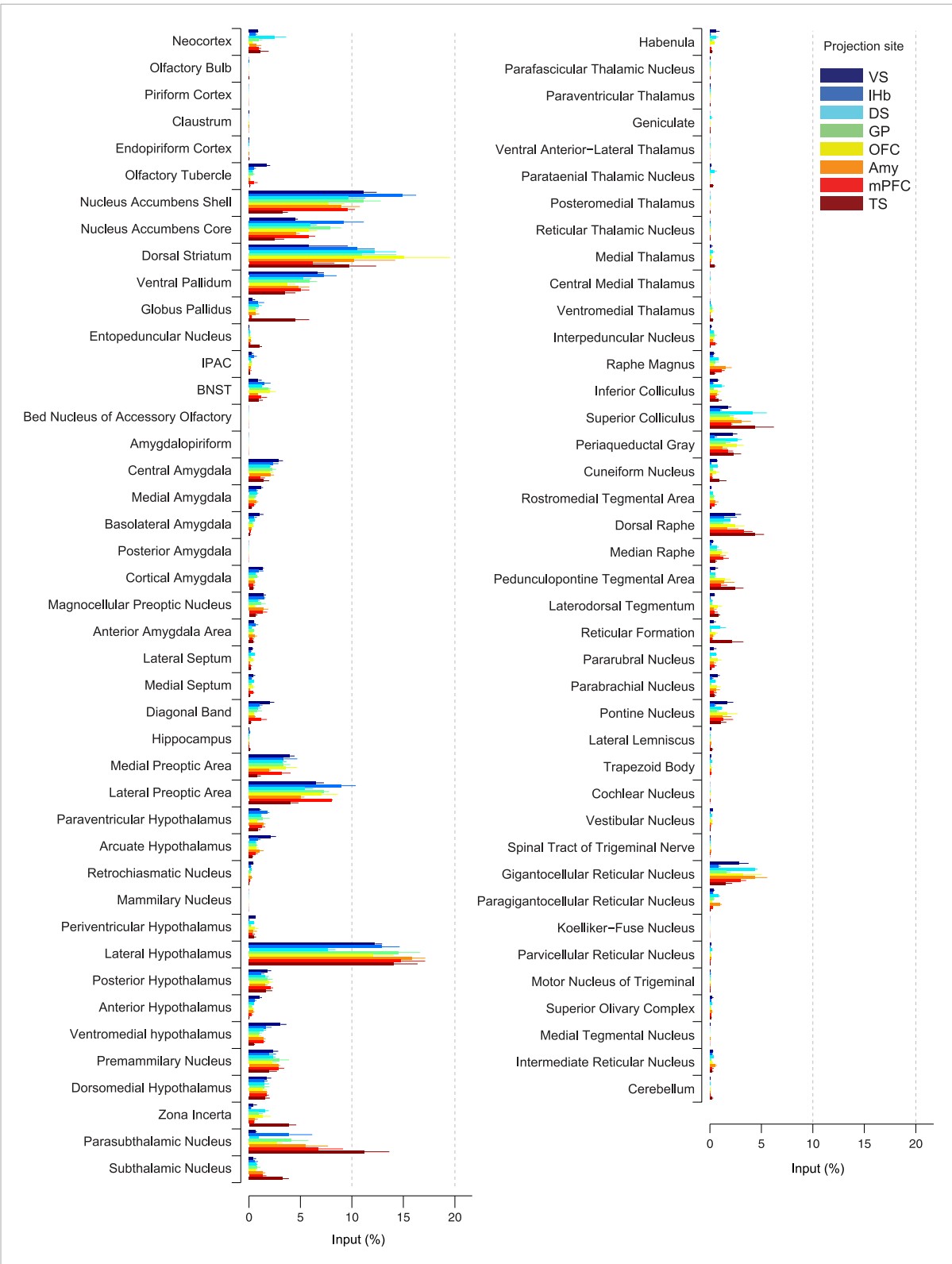

**Figure 4**. Comparison of the percentage of inputs from each region across populations of projection-specific dopamine neurons. A summary of the distribution of inputs to dopamine neurons with different projection sites, with bars representing the average % of inputs observed (out of all labeled input neurons outside of the VTA/SNc/SNr/RRF) per region (mean ± s.e.m.). Each color represents inputs to a different population of dopamine neurons, as indicated in the inset. The 20 most prominent inputs are compared in *Figure 4—figure supplement 1*.

*Figure 4. continued on next page*

*Figure 4. Continued*

The following figure supplement is available for figure 4:

**Figure supplement 1**. Comparison of the percentage of inputs from selected regions across populations of projection-specific dopamine neurons.

similar to VS-projecting dopamine neurons (which are known to encode RPE), while TS-projecting dopamine neurons have a unique distribution of inputs and therefore form a distinct class of dopamine neurons.

By carefully examining the data, we could also discern some trends among the 'canonical' populations of dopamine neurons. For example, DS-projecting dopamine neurons received more inputs from the neocortex than other populations (*Figure 4*), while VS-projecting dopamine neurons received more inputs from the olfactory tubercle and other ventromedial structures such as the ventromedial hypothalamus (*Figure 4*). Furthermore, DS-projecting and OFC-projecting dopamine neurons appeared to receive more inputs from the dorsal-most part of the DS (*Figure 3B*). In the posterior midbrain, all populations received inputs from the dorsal raphe, but lHb-projecting dopamine neurons received fewer inputs from nearby areas (*Figure 3H*). In addition, lHb-projecting dopamine neurons received fewer inputs from posterior midbrain regions such as the periaqueductal grey and gigantocellular reticular nuclei (*Figure 4*). By far, though, the most distinct outlier was the set of inputs to TS-projecting dopamine neurons. In all of the cases in which our data attained statistical significance across all populations after correction for multiple comparisons (ventral striatum, ventromedial hypothalamus, GP, ZI, and STh), TS-projecting neurons were the clear outliers (*Figure 4—figure supplement 1*).

To further quantify the overall similarity of patterns of inputs between conditions, we calculated pair-wise correlations (Pearson's correlation coefficients) between the percentage of input neurons across anatomical areas (*Figure 5*). We found that most pairs had relatively high correlations ($r = 0.85$–$0.98$). Here again, TS-projecting dopamine neurons had consistently lower correlations than other populations (*Figure 5A*): TS-projecting dopamine neurons were most distinct from VS- and DS-projecting dopamine neurons ($r = 0.65$ and $0.69$, respectively) (*Figure 5A*), although the similarity was slightly higher compared to non-striatal regions ($r = 0.69$–$0.82$). Hierarchical clustering supported this view: TS-projecting dopamine neurons were an outlier among the populations tested (*Figure 5B*). Together, these results show that TS-projecting dopamine neurons receive the most unique set of inputs among the populations of dopamine neurons that we examined.

## Inputs to TS-projecting dopamine neurons are dorso-laterally shifted compared to inputs to other populations of dopamine neurons

When parceling the data based on brain regions, we found that most dopamine neurons receive more inputs from structures along the ventral medial part of the basal ganglia and hypothalamus, while TS-projecting dopamine neurons receive more inputs from subthalamic areas. Because we noticed that labeled neurons often did not respect the boundaries between regions (i.e., they did not form discrete clusters corresponding to each sub-region) and that distributions of labeled neurons were not always uniform in each area (e.g., the LH), we also examined the general topography of inputs in these areas irrespective of regional boundaries.

We found that, in several coronal planes along the A-P axis, the center of mass of labeled neurons differed between conditions. Interestingly, we observed that the inputs to TS-projecting dopamine neurons were shifted dorso-laterally compared to the inputs to other subpopulations (*Figure 6*). In the anterior part of the brain, this shift is caused by the relatively low number of inputs from the ventral striatum to TS-projecting dopamine neurons (*Figure 6A*). In the middle sections of the brain, the shift is caused by an increased number of inputs from GP (*Figure 6B*) and EP (*Figure 6C*), as well as a decrease in inputs from the ventromedial hypothalamus. Finally, in a more posterior section of the brain, this shift is caused by the large number of inputs from the subthalamic and parasubthalamic nuclei to TS-projecting neurons (*Figure 6D*), as well as the lower number of inputs from the ventromedial hypothalamus.

In summary, we found that the center of mass of inputs to TS-projecting dopamine neurons were dorso-laterally shifted compared to inputs to other subpopulations in several coronal planes (*Figure 6E–H*). This suggests that the input pattern to TS-projecting dopamine neurons was unique,

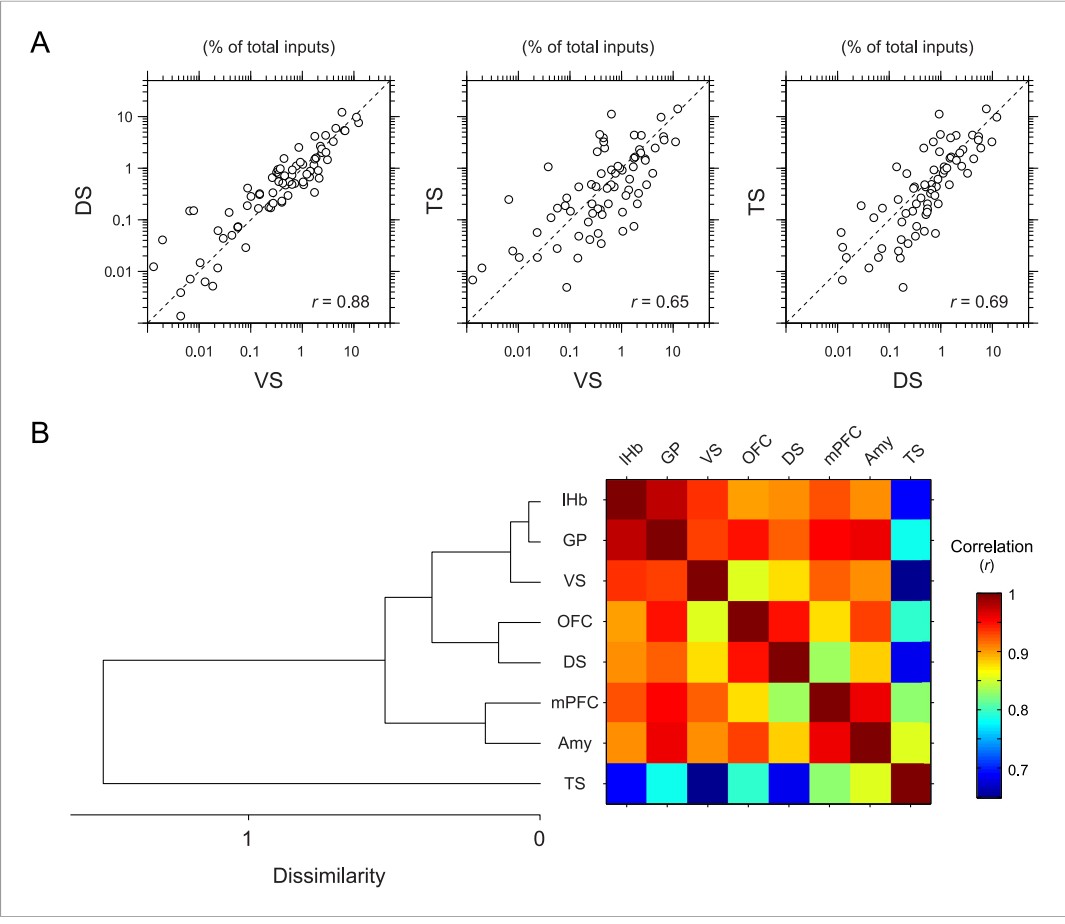

**Figure 5**. Correlation between projection-specific populations of dopamine neurons. (**A**) Scatter plots comparing the percent of inputs from each anatomical region for VS-, DS-, and TS-projecting dopamine neurons. Each circle represents the average percent of inputs for one anatomical area for each condition. *r*: Pearson's correlation coefficient. The axes are in log scale. (**B**) Dendrogram (left) and correlation matrix (right) summarizing all pair-wise comparisons. Color on the right indicates correlation values. The dendrogram was generated based on hierarchical clustering using the average linkage function.

regardless of the method used for analysis. Although the meaning of the shift is not clear, overall, our analysis revealed that the inputs to subpopulations of dopamine neurons were topographically organized within and across nucleus boundaries.

## TS-projecting dopamine neurons do not receive inputs from 'clusters' within the VS that are common sources of inputs to other dopamine neurons

As we discussed above, the most prominent inputs to the 'canonical' dopamine neurons are from the nucleus accumbens, a part of the ventral striatum, though TS-projecting dopamine neurons received far fewer inputs from the nucleus accumbens (*Figure 3A*; *Figure 4*, *Figure 7A–D*). For each of the 'canonical' populations, between 12% and 20% of the input neurons were located in the ventral striatum whereas only 5% of the input neurons were located in the ventral striatum for TS-projecting dopamine neurons (*Figure 7D*).

Our labeling showed discrete peaks of high density within the nucleus accumbens that do not correspond to the boundaries of commonly delineated sub-areas of this region (i.e., 'core' or 'shell') (*Figure 7E–H*). Our previous study reported the existence of these 'ventral patches' in consistent locations across animals (*Watabe-Uchida et al., 2012*). In the present study, using precise 3D density plotting, we determined the exact number and locations of these patches: 5 total, with 3 in the

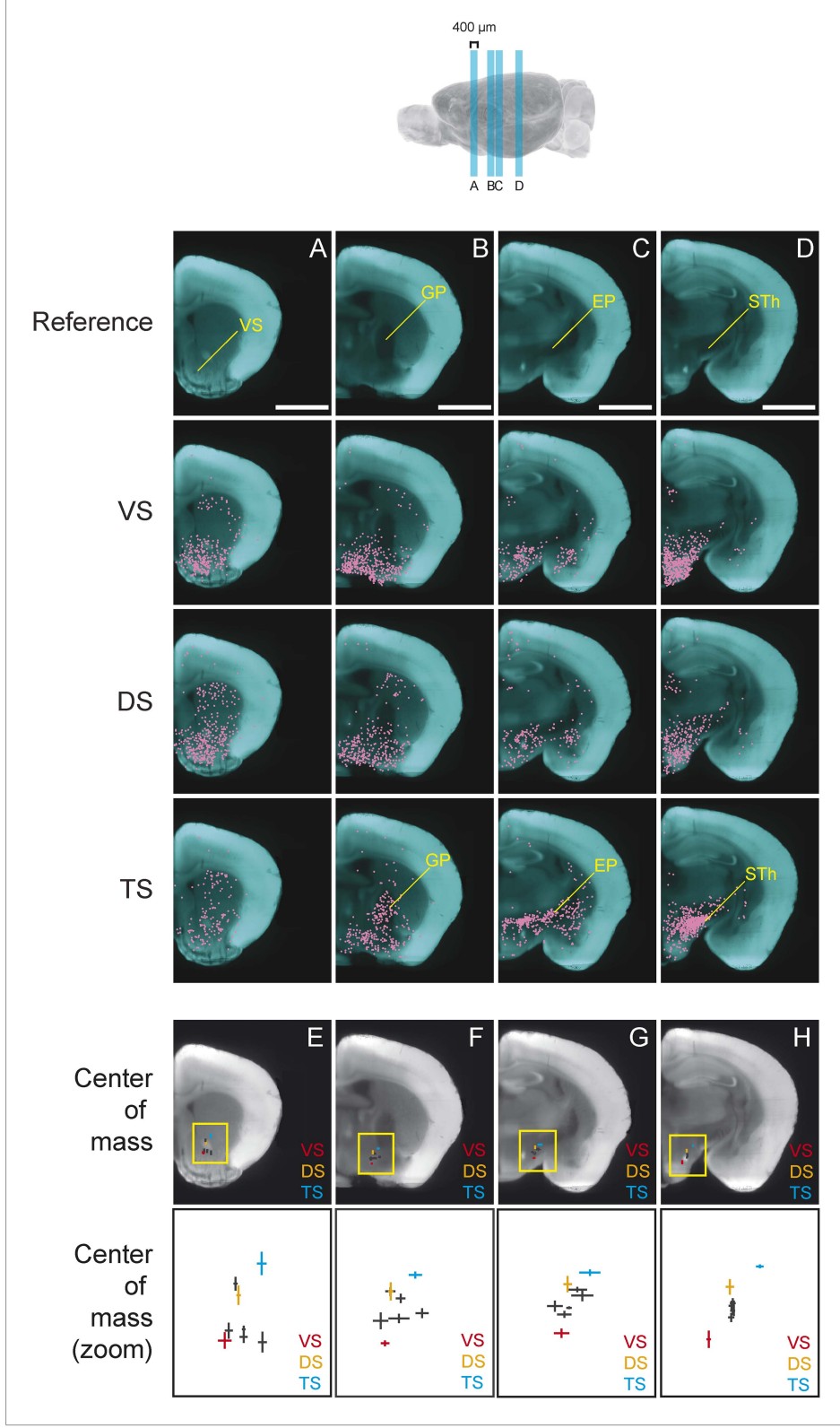

**Figure 6**. Topological shift in the center of mass of input neurons to projection-specified dopamine neurons. Four coronal optical sections (400 µm thick) were chosen to demonstrate the dorsolateral shift of inputs to TS-projecting dopamine neurons. (**A–D**) Coronal optical sections, with a region of interest marked in yellow for reference. Bars represent 2 mm. The distribution of inputs to each population of neurons is plotted in magenta and neurons within

*Figure 6. Continued*

the ventral striatum (**A**), GP (**B**), entopeduncular nucleus (EP) (**C**), and subthalamic nucleus (STh) (**D**) are indicated. A fixed number of neurons were randomly chosen from each brain and those neurons from three randomly chosen animals per condition were plotted for corresponding coronal sections. (**E**–**H**) Coronal optical sections, with a yellow box showing the location of the insets displayed below. The center of mass for each population is shown, with vertical and horizontal lines representing the standard error in the y-axis and x-axis, respectively.

anterior medial ventral shell, 1 in the medial core and 1 in the posterior lateral shell (*Figure 7G,H*, *Figure 7—figure supplement 1*).

We observed these dense patches of inputs in all cases, with the exception of inputs to TS-projecting dopamine neurons (*Figure 7A–C,J*). Of the input neurons in the nucleus accumbens, 16–26% were localized to these patches for all of the 'canonical' populations (*Figure 7I*). By contrast, the few inputs that TS-projecting neurons did receive from the nucleus accumbens were not concentrated in patches (*Figure 7C,I,J*). Thus, whereas neurons in the ventral patches may play a key role in regulation of dopamine neurons, TS-projecting dopamine neurons may be relatively disconnected from this regulatory system.

We next compared the distribution of inputs from these patches between different populations of dopamine neurons. We found that each subpopulation received inputs differentially from different patches. For example, mPFC-projecting dopamine neurons received a larger ratio of inputs from the most medial patch, while Amy-projecting dopamine neurons received more of their inputs from the lateral patches (*Figure 7J*). Each of these patches could potentially have a distinct functional role, and their contributions to dopamine neurons should be considered individually in future experiments.

## Reciprocity is not a defining characteristic of midbrain dopamine neuron circuitry

We observed that most populations of dopamine neurons receive a large number of monosynaptic inputs from the ventral striatum. Because dopamine neurons also innervate the ventral striatum, we wondered whether VS-projecting neurons receive a larger proportion of inputs from the VS than other populations and whether the other populations received a larger proportion of inputs from their respective projection sites. Because we labeled the AAV-FLEX-TVA virus injection sites with BFP (to label the areas that the infected dopamine neurons project to, in each case), monosynaptic inputs residing within the BFP-labeled regions are more likely to have reciprocal connections with dopamine neurons. Because the CLARITY-based clearing process completely degraded the native fluorescence of BFP, we sliced the brains into 1 mm sections and stained them with an anti-BFP antibody to reveal the injection sites. We examined the distribution of inputs with respect to these injection sites. Surprisingly, we did not observe that monosynaptic inputs accumulated in BFP-labeled areas (*Figure 8A–F*).

To quantify this observation, we compared the number of potential reciprocal inputs to each subpopulation to the average number of inputs in this area among all other subpopulations. First, we compared how many inputs each population receives from the nucleus that it projects to. We found that there was some variability, but that no population received a statistically significantly higher fraction of inputs from the region that it projected to (*Figure 8G*). Because some of our injections were limited to a portion of a region (for example, our injection into the DS only covered the anterior medial DS), we did a further analysis using the injection sites instead of the targeted nuclei to quantify the data. With this analysis, we also found no statistically significant differences (*Figure 8H*). In short, we observed that reciprocal monosynaptic connections are not a defining characteristic of inputs to dopamine neurons with differing projection targets (*Figure 9*). Rather, we found that most dopamine neurons (besides TS-projecting dopamine neurons) have similar inputs regardless of their projection target.

## Discussion

In this study, we mapped the monosynaptic inputs to subpopulations of dopamine neurons defined by their projection targets. Our data show that reciprocal connectivity with target areas is not

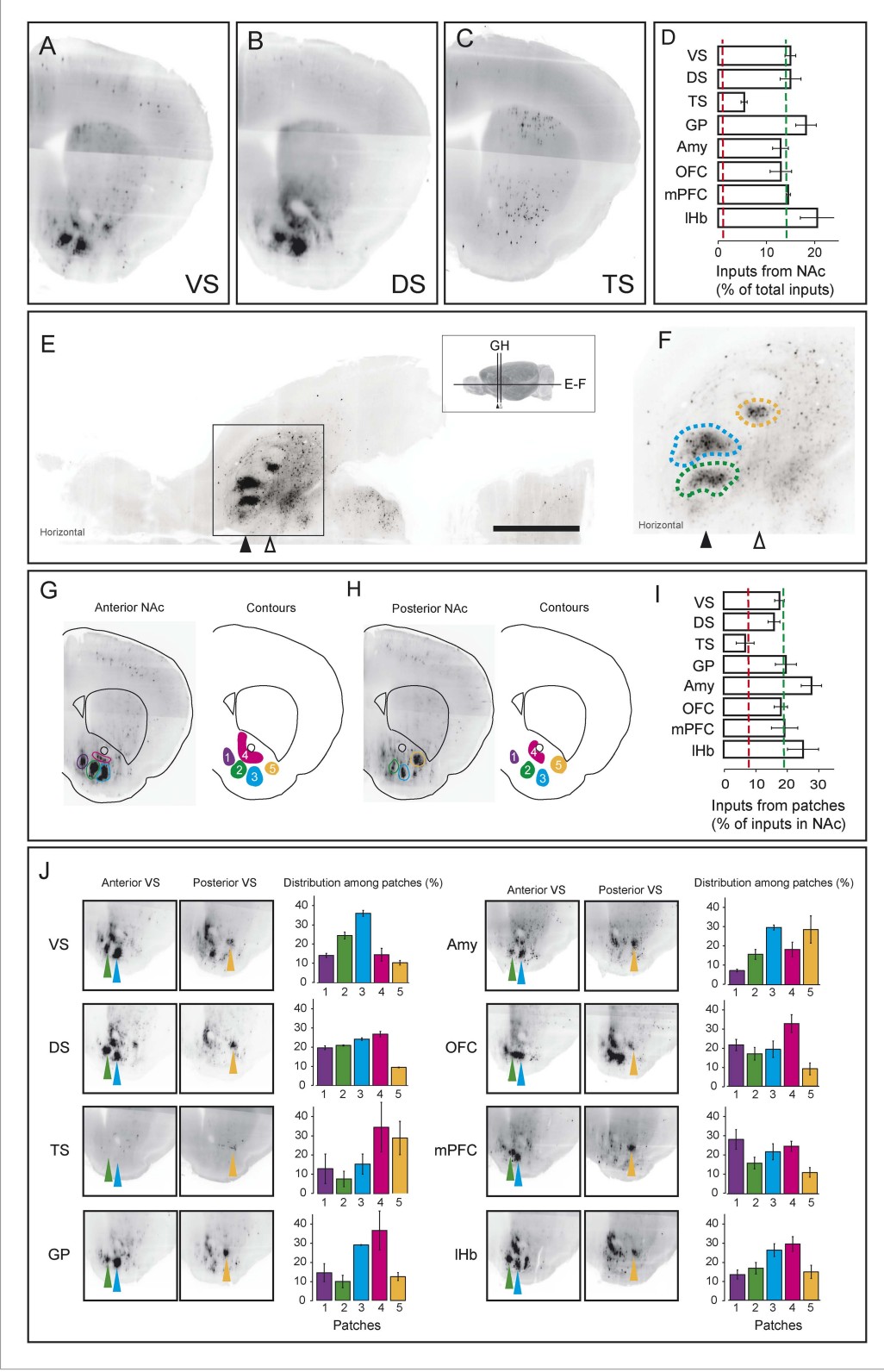

**Figure 7**. Dense regions of inputs within the ventral striatum. (**A–C**) Coronal optical sections showing typical distributions of inputs in the ventral striatum. A comparison of the typical 'patch' structure of inputs from the ventral striatum (to VS-projecting or DS-projecting dopamine neurons, for example) with the atypical 'patch-less' structure of inputs from the striatum to TS-projecting dopamine neurons. (**D**) Percentage of input neurons within the ventral

*Figure 7. continued on next page*

*Figure 7. Continued*

striatum (core, medial shell, and lateral shell combined) in each condition. Red dotted line indicates the percentage expected based on chance distribution throughout the brain, while green dotted line indicates the average percentage among all animals. (**E**) Horizontal optical section showing a typical distribution of inputs in the ventral striatum, with many input neurons in tight clusters. Bar represents 2 mm. (**F**) A zoomed view of **E**, taken from the indicated box. Clusters are dispersed along the A-P axis, so two planes are used to display them in coronal sections: the black arrow indicates the anterior plane and the white arrow indicated the posterior plane. (**G–H**) Coronal optical sections showing the two planes indicated above, as well as a graphical representation of the five patches in those planes. (**I**) Among labeled neurons in the ventral striatum, the percentage within the patches in each condition. Red dotted line indicates the percentage expected based on chance distribution throughout the nucleus accumbens, while green dotted line indicates the average percentage among all animals. (**J**) Among labeled neurons within the patches, the percentages within each patch in each condition. Green arrows point to patch 2, cyan arrows point to patch 3, and yellow arrows point to patch 5. More detailed description of the patches shown in ***Figure 7—figure supplement 1***.

The following figure supplement is available for figure 7:

**Figure supplement 1**. Density-based analysis of inputs to projection-specified dopamine neurons.

a prominent feature of dopamine neurons. Instead, most populations of dopamine neurons receive a surprisingly similar set of inputs. On the other hand, we found a systematic outlier: dopamine neurons projecting to the tail of the striatum receive a distinct set of inputs compared to dopamine neurons projecting to other targets. This conclusion was supported by the following observations. First, when the percent of inputs from a given area was found significantly different among conditions (1-way ANOVA), TS-projecting dopamine neurons were the main driver of these differences. TS-projecting dopamine neurons received far fewer inputs from the nucleus accumbens, whereas TS-projecting neurons received uniquely higher numbers of inputs from the GP, STh, and ZI. Second, our examination of the spatial shift of input neurons in coronal optical sections also indicated that the center of mass of inputs to the TS-projecting population tended to be located more lateral and dorsal while the center of mass of inputs for the VS-projecting population tended towards the opposite. Third, correlation analysis and hierarchical clustering indicated that the pattern of inputs to TS-projecting dopamine neurons was most distinct from other conditions. Together, these results indicate that the dopamine neurons projecting to the posterior striatum form a distinct class of dopamine neurons.

## The basic architecture of the dopamine circuit

The brain performs complicated tasks using layers of hierarchical and/or parallel neural circuits. Alheid and Heimer proposed that a major organizational principle of the brain is the existence of parallel functional and anatomical macrosystems, each comprised of a cortical area, a cortical input nucleus (such as an area in the striatum), an output nucleus (such as a part of the pallidum), thalamus, and brainstem (*Alheid and Heimer, 1988*; *Zahm, 2006*). According to this view, each macrosystem (such as the ventral striatum, the DS, or the extended amygdala) would have its own dedicated dopamine system, resulting in a circuit organization resembling many parallel closed loops. This idea is plausible, considering the topographical organization of the dopaminergic nuclei (*Swanson, 1982*; *Yetnikoff et al., 2014*), and it could explain the diverse function of dopamine neurons. If dopamine neurons encode RPE, each macrosystem could learn independently through its own feedback. To test this directly, we asked whether dopamine neurons projecting to a nucleus in one macrosystem receive a larger proportion of their inputs from that nucleus or from other nuclei in that same macrosystem.

In stark contrast to this proposal, we found only loose reciprocity between dopamine neurons and their projection targets. If RPE is used to correct a neuron's behavior so that the neuron could represent predicted value more accurately in the future, a logical structure could be for prediction (expectation) to be sent to dopamine neurons and then for the same dopamine neurons to send back the error of the prediction (RPE) to the same neuron or the same system. Although many areas that receive dopamine projections also send projections to dopamine neurons, we found that reciprocal connections are not a basic principle explaining the distribution of monosynaptic inputs. In other words, reciprocal connectivity might not be the general mechanism of RPE computation and

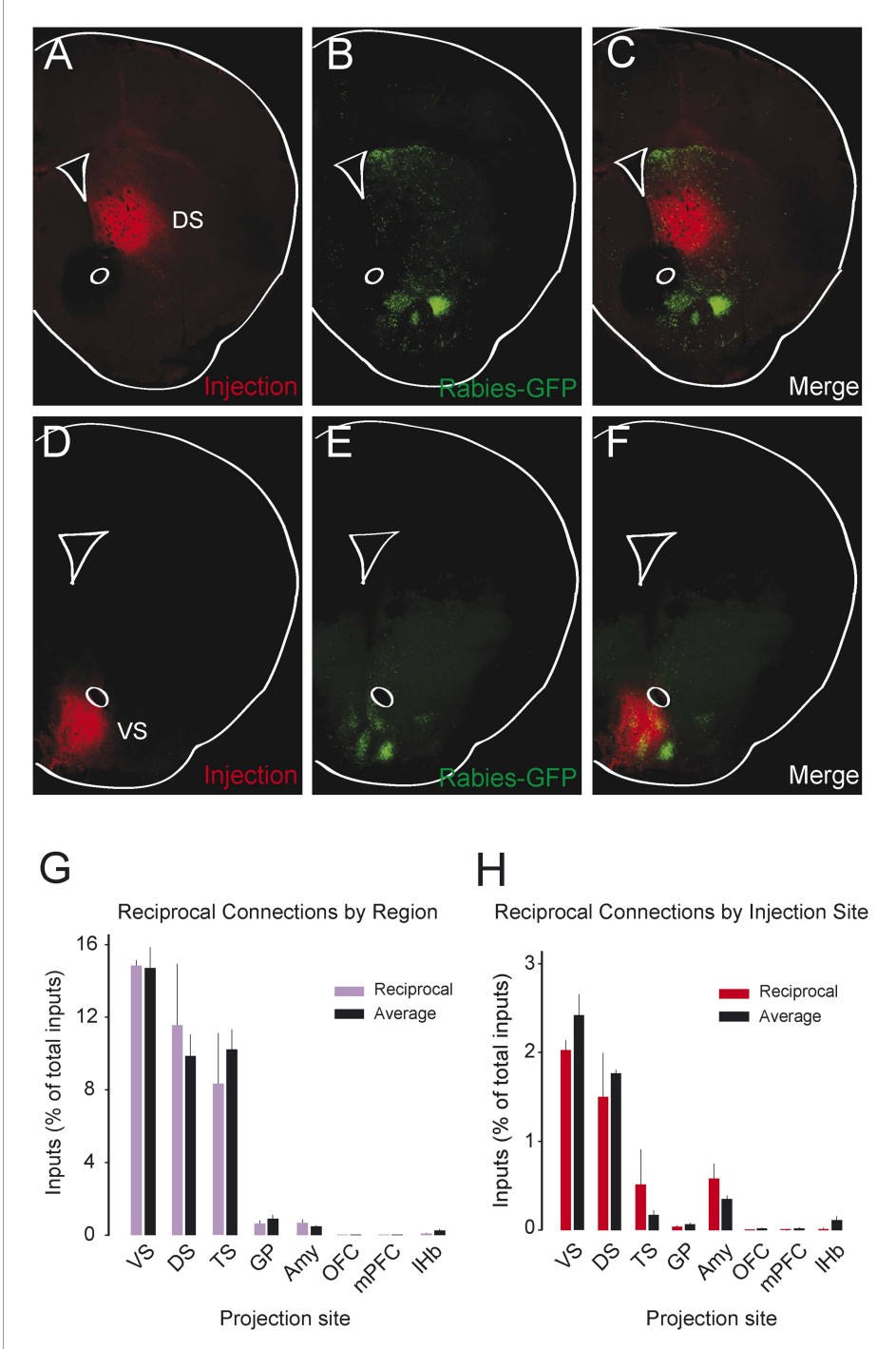

**Figure 8**. Reciprocity of connection between dopamine neurons and neurons at their projection sites. CLARITY-based brain clearing did not preserve native BFP fluorescence, so brains were physically sectioned and stained with an anti-BFP antibody to label the injection site of AAV-FLEX-TVA in each animal. (**A–C**) The injection site (red) and input neurons (green) labeled in a physical coronal section for DS-projecting dopamine neurons. (**D–F**) The injection site (red) and input neurons (green) labeled in a physical coronal section for VS-projecting dopamine neurons.
(**G**) A comparison of the percentage of inputs from the reciprocal region (defined by brain region) of each injection site (in purple) with the average percentage of inputs from that region among all other brains (in black). For this analysis, the DS was split into 'anterior DS' for DS and 'posterior DS' for TS at Bregma −0.9 mm. There were no significant differences between pairs (two-sample t-test). (**H**) A comparison of the percentage of inputs from the reciprocal site (defined by region of BFP infection) of each injection (in red) with the average percentage of inputs from that region among all other brains (in black). There were no significant differences between pairs (two-sample t-test).

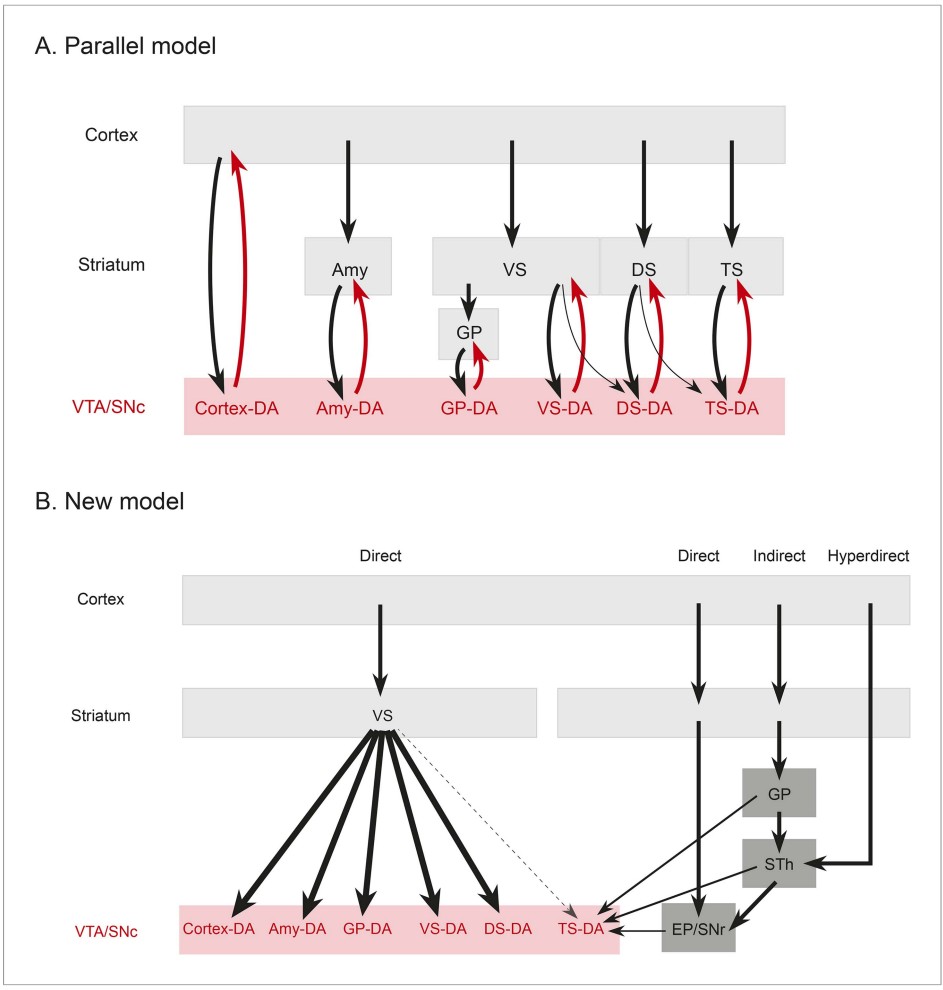

**Figure 9**. Summary and model. (**A**) A working model suggesting that dopamine neurons receive inputs primarily from the region that they project to, based on the idea that input neurons provide an 'expectation' signal and dopamine neurons correct them (i.e., send them a prediction error signal based on the type of expectation that they encoded) individually. (**B**) Our model, in which dopamine neurons in the 'canonical' pathway receive inputs primarily from a common set of regions (and particularly heavy inputs from the ventral striatum) and send a common prediction error signal to many parts of the forebrain. We propose that TS-projecting dopamine neurons could be part of a relatively separate pathway with a potentially unique function (different from reward prediction error (RPE) calculation) based on its unique distribution of inputs. Some common inputs such as the DS and ventral pallidum are omitted for clarity.

dopamine functions. Our data suggests that neurons in cortex, striatum, pallidum, habenula, or amygdala may not 'entrain' different populations of dopamine neurons. Rather, most dopamine neurons are more likely to gather information from a set of common sources and broadcast a common RPE to many brain areas (*Figure 9*), while specific populations with different sets of inputs, such as TS-projecting dopamine neurons, might have different functions.

Previously, we found that many brain regions send monosynaptic inputs to dopamine neurons (*Watabe-Uchida et al., 2012*). However, which of these brain areas are most important for RPE computation remains unclear. Because the ventral striatum was reported to receive a RPE signal from dopamine neurons (*Roitman et al., 2008*; *Clark et al., 2012*; *Hart et al., 2014*), we reasoned that finding the monosynaptic inputs to VS-projecting dopamine neurons would narrow down the list of inputs that are potentially important to RPE computation. Although we were able to rule out some of the areas mentioned above which preferentially project to TS-projecting dopamine neurons, many brain areas still remain on the list of prominent inputs to VS-projecting dopamine neurons. Based on this logic, many brain areas have the potential to directly modulate RPE in dopamine neurons.

## The tail of the striatum

Our results suggest that dopamine neurons projecting to the anterior vs posterior part of the striatum receive distinct inputs. Considering that dopamine plays essential roles in striatal functions, this result provides insight into the functioning of different parts of the striatum. Most commonly, the striatum is divided into two parts along the dorso–ventral axis: ventral striatum (or nucleus accumbens in rodents, also called the 'limbic' striatum) and DS. In addition, the striatum is divided into more sub-regions by splitting the ventral striatum into 'core' and 'shell' regions and splitting the DS into dorso-medial (associative) and dorso-lateral (motor-related) regions (*Francois et al., 1994*; *Lynd-Balta and Haber, 1994*; *Haber et al., 2000*; *Joel and Weiner, 2000*; *Redgrave et al., 2010*). In primates, projections from different cortical areas define different striatal regions. Projections from dorsolateral prefrontal cortex (DLPFC) and pre-supplementary motor cortex preferentially target associative striatum, whereas projections from OFC are unique to the limbic striatum. In rodents, there is a substantial controversy over the existence and location of homologous structures to the DLPFC or OFC of the monkey (*Preuss, 1995*; *Uylings et al., 2003*; *Vertes, 2004*; *Kita et al., 2014*). Thus, it is difficult to define the subareas of the striatum in the mouse brain. Nonetheless, even if it is not perfectly homologous to the primate striatum, previous studies suggest that the rodent striatum could contain a functional gradient from ventromedial to dorsolateral in agreement with the organization of cortical projections (*Berendse et al., 1992*; *Voorn et al., 2004*).

The three regions of the striatum that we targeted for investigation, VS, DS, and TS may reside roughly in this order in this gradient (*Figure 3—figure supplement 5*). Our target for DS (or the dorsal head) belongs to the 'dorso-medial' subarea, whereas TS (or the posterior striatum) consists of proportionally a more 'dorso-lateral' portion with some 'dorso-medial' and 'ventral' portions. Supporting the notion that these populations form a functional gradient, the input patterns to TS-projecting dopamine neurons are slightly more similar to that of DS-projecting dopamine than VS-projecting dopamine (*Figure 8*). However, we found that the similarity of inputs between DS- and VS- projecting dopamine neurons was much higher than similarity of inputs between DS- and TS- projecting dopamine neurons. This could potentially be explained by the fact that our injections into DS and VS were much closer together along the A-P axis (*Figure 3—figure supplement 5*). It may be fruitful to investigate a potential functional gradient along the A-P axis of the striatum in the future.

Previous studies suggested that there is a functional difference between anterior and posterior striatum in monkey and human (*Miyachi et al., 1997*; *Lehericy et al., 2005*). For example, it has been shown that the anterior striatum is important for skill learning and the posterior part is important for skill execution in monkeys (*Miyachi et al., 1997*). More recently, Kim and Hikosaka showed that the anterior caudate encodes flexible values, whereas the posterior caudate encodes stable values (*Kim and Hikosaka, 2013*). It would be interesting to determine whether this type of specialization for flexible or stable value coding applies also to the anterior and posterior parts of the putamen. It also remains unknown whether the striatum in rodents contains similar functional differences between the anterior and posterior parts.

Increasing evidence suggests that dopamine neurons projecting to the ventral striatum signal RPE (*Roitman et al., 2008*; *Clark et al., 2012*; *Hart et al., 2014*), however, it remains to be clarified whether dopamine neurons projecting to other brain areas send distinct signals. Our results raise the possibility that dopamine neurons projecting to the tail of the striatum convey a different type of information than RPE. One interesting possibility is that dopamine signals in the tail of the striatum lack mechanisms to update the value representation based on the consequences of actions, consistent with the idea that the dopamine recipient neurons residing in this part of the striatum store 'stable values' (*Kim and Hikosaka, 2013*). Dopamine is involved in different aspects of reward-based learning (e.g., goal-directed vs habit) as well as a wider range of functions besides reward-based learning, including motor and cognitive functions. It is therefore possible that TS-projecting dopamine neurons are related to habitual behavior or one of these other functions, unlike other dopamine neurons that signal RPE.

## Unique set of monosynaptic inputs to TS-projecting dopamine neurons

The most prominent feature of TS-projecting dopamine neurons is that there are many fewer inputs from the ventral striatum (especially from the ventral patches), whereas the ventral striatum is one of the largest sources of inputs to the other dopamine neurons we investigated. Various models of RPE

computations assume specific roles for inputs from particular areas. For instance, it has often been assumed that the striatum sends inhibitory 'expectation' signals to dopamine neurons that can 'cancel out' reward values when they are expected (*Houk and Adams, 1995*; *Kobayashi and Okada, 2007*; *Doya, 2008*; *Rao, 2010*; *Khamassi and Humphries, 2012*). Our finding that most populations of dopamine neurons received heavy innervation from the ventral striatum, which might encode 'expectation' or 'state values' in a RPE calculation, suggest that dopamine signals in these target areas are suppressed when reward is expected. On the other hand, our results indicate that TS-projecting dopamine neurons may not be subject to this type of regulation.

Rather than the ventral striatum, TS-projecting dopamine neurons received more inputs from the STh, PSTh, and ZI. STh is known for its role in motor function and is a uniquely glutamatergic structure of the basal ganglia (whereas all the other nuclei in basal ganglia are mainly inhibitory). STh receives a strong projection from the GP in addition to the cortex (*Canteras et al., 1990*; *Francois et al., 2004*; *Kita et al., 2014*) and sends projections to other basal ganglia nuclei, such as striatum, GP, EP and SNr (*Kita and Kitai, 1987*; *Degos et al., 2008*). Interestingly, TS-projecting dopamine neurons receive inputs from STh as well as from areas that tend to be heavily interconnected with STh, such as GP, EP and SNr. These results suggest that, instead of inhibitory inputs from the ventral striatum ('direct pathway'), TS-projecting dopamine neurons receive more glutamatergic inputs from the STh ('hyperdirect pathway'), inhibitory inputs from pallidum ('indirect pathway'), and also inputs from two output nuclei: SNr and EP (*Figure 9*). Thus, cortical information may be transmitted to TS-projecting dopamine neurons through different pathways than other dopamine neurons. Finally, ZI, which is related to arousal, and PSTh, which is related to autonomic function, (*Goto and Swanson, 2004*; *Kita et al., 2014*), are additional unique pathways from cortex to TS-projecting dopamine neurons.

## Ventral patches

Previous studies have shown that there are functional and histological subdivisions of the ventral striatum (*Zahm and Brog, 1992*; *Pennartz et al., 1994*). However, to our knowledge, there is no clear structure or function corresponding to the ventral patches that we described. In a previous study, we found that calbindin, a marker for patches in the DS, partially distinguished the ventral patches from surrounding areas (*Watabe-Uchida et al., 2012*). These patches are, in essence, dense 'hot spots' for inhibitory inputs to dopamine neurons whereas neurons outside the patches are predominantly indirectly (if at all) connected to dopamine neurons. One interesting possibility is that these ventral patches are negatively or positively correlated with the observation of 'hedonic hot spots' within the striatum. These hot spots are so named because the 'hedonic' desire for food was enhanced by the local stimulation of μ-opioid receptors residing in the medial-dorsal part of the anterior ventral striatum but not by the stimulation of neighboring areas (*Castro and Berridge, 2014*). Overall, the type of information each part of the striatum relays to dopamine neurons is an open question. Our anatomical definition of the stereotypical locations of the ventral patches will help guide future studies of what information is channeled through these microdomains (ventral patches) of the ventral striatum.

## Relationship to Parkinson's disease

In Parkinson's disease, dopamine neurons in the SNc are progressively lost as the disease worsens. In particular, there is a prominent loss of dopaminergic axons in the posterior putamen and a corresponding loss of dopamine neurons in the ventrolateral SNc (*Kish et al., 1988*; *Fearnley and Lees, 1991*; *Frost et al., 1993*; *Morrish et al., 1995*; *Damier et al., 1999*; *Pavese and Brooks, 2009*; *Redgrave et al., 2010*). Interestingly, dopamine neurons projecting to the tail of the striatum preferentially reside in the lateral part of SNc, suggesting that the tail of the striatum in mice may be homologous to the posterior putamen.

One prominent feature of TS-projecting dopamine neurons is relatively strong input from STh. STh is uniquely excitatory compared to the other nuclei of the basal ganglia. Although the etiology of Parkinson's disease remains unclear, it has been proposed that uncontrolled, excessive excitation may eventually kill dopamine neurons (*Rodriguez et al., 1998*; *Olanow and Tatton, 1999*). The strong monosynaptic excitation from STh could at least partially explain why dopamine neurons that project to the posterior putamen are specifically vulnerable.

Because of the side effects of systemic L-DOPA administration, more and more patients have been receiving deep brain stimulation (DBS) as a treatment for Parkinson's disease (*Benabid et al., 2009*;

*Chaudhuri and Odin, 2010*; *Cyron et al., 2010*; *Deniau et al., 2010*; *Ponce and Lozano, 2010*). Currently, the main targets for DBS are STh, ZI, GPi (also called EP in rodents) and PPTg. Interestingly, our results showed that TS-projecting dopamine neurons receive monosynaptic inputs preferentially from all of these areas. In other words, effective DBS sites might reside in presynaptic sites that preferentially target TS-projecting dopamine neurons, rather than common inputs to all the dopamine neurons such as LH. As a result, DBS applied to these areas might influence only the circuits relevant to the TS-projecting dopamine system (most of which is lost in patients) without changing other aspects of dopamine functions, such as learning and reward-seeking behaviors. If this is true, the list of monosynaptic inputs to TS-projecting dopamine neurons will be a useful reference for seeking new DBS targets. Furthermore, expanding the framework for understanding the mechanism of DBS function could help improve its efficiency.

## Technical considerations and relation with recent studies

In this study, we examined dopamine subpopulations that project to eight different projection targets. Two studies were published very recently that are highly related to our present work. One study found that monosynaptic inputs to dopamine neurons in VTA that project to medial and lateral NAc, mPFC and Amy were similar in all the areas they examined, with the exception of inputs from the dorsal raphe and within NAc (*Beier et al., 2015*). Similarly, another study found that monosynaptic inputs to neurons in SNc that project to medial DS and lateral DS were similar with the exception of inputs within DS (*Lerner et al., 2015*). Although neither of these studies examined TS-projecting dopamine neurons, the results are largely consistent with our finding that monosynaptic input patterns to most dopamine subpopulation are similar, and strengthen our finding that TS-projecting dopamine neurons are relatively unique. However, the patterns of inputs for VS-projecting neurons shown in the former study (*Beier et al., 2015*) and those for DS-projecting dopamine neurons in the latter study (*Lerner et al., 2015*) were quite different, in contrast to our finding. We previously observed different patterns of input between VTA and SNc dopamine neurons such as a strong preferential projection from DS to SNc (*Watabe-Uchida et al., 2012*). In the present study, we found that most populations of projection-specific dopamine neurons are distributed both in VTA and SNc. We, therefore, injected rabies virus both in VTA and SNc for all eight conditions. The large difference in inputs between VS-projecting VTA neurons (*Beier et al., 2015*) and DS-projecting SNc populations (*Lerner et al., 2015*) may not be due to the difference in projection targets (VS vs DS), but rather because of the location of their virus injection sites (VTA vs SNc). It remains to be examined whether VTA dopamine neurons with a specific projection site receive a different set of monosynaptic inputs compared to SNc dopamine neurons that project to the same site (i.e., VTA → DS neurons vs SNc → DS neurons).

Our study relies on the assumption that each dopamine subpopulation defined by projection target is a different population. In general, dopamine axons from both VTA and SNc are mainly unbranched and thus target one structure (*Fallon et al., 1978*; *Swanson, 1982*; *Sobel and Corbett, 1984*; *Febvret et al., 1991*; *Matsuda et al., 2009*). However, there is some disagreement about the extent of collateralization of dopamine axons (*Yetnikoff et al., 2014*); some dopamine neurons may project to multiple brain areas (*Fallon and Loughlin, 1982*; *Loughlin and Fallon, 1984*; *Takada and Hattori, 1986*). Using CLARITY and whole brain imaging with membrane-bound GFP, Lerner et al. found that DS-projecting SNc neurons have negligible collateralization to NAc and no collateralization to other dopamine target areas such as prefrontal cortex or amygdala (*Lerner et al., 2015*). On the other hand, using conventional light microscope, Beier et al. found that lateral NAc-projecting dopamine neurons have broad arborizations going to DS, whereas medial NAc-projecting dopamine neurons do not project to the DS (*Beier et al., 2015*). They found that neither dopamine groups collateralize to mPFC or amygdala (*Beier et al., 2015*). In spite of some discrepancies between studies, these studies show that dopamine neurons projecting to the target areas which we chose in the present study consist of largely non-overlapping subpopulations. However, because these studies could have overlooked sparse axon signals, we must interpret the results carefully, because any populations of dopamine neurons infected due to their projections to multiple areas would make the data appear more similar.

Our viral tracing method may allow trans-synaptic spread between dopamine neurons, which would make the trans-synaptically labeled populations appear more similar. However, the intrinsic connections in VTA/SNc originate mainly from non-dopamine neurons without TH immunoreactivity

(*Bayer and Pickel, 1990*; *Ferreira et al., 2008*; *Jhou et al., 2009*). With respect to potential trans-synaptic spread, aforementioned studies overcame this problem by using CAV-Cre in wild type mice, sacrificing cell type specificity (*Lerner et al., 2015*) or by establishing a technically advanced method using CAV virus combined with Cre and Flp recombinase (*Beier et al., 2015*). Importantly, these studies also found that subpopulations of dopamine neurons had largely similar monosynaptic inputs.

We found that dopamine neurons that were infected with AAV5-FLEX-TVA at axon terminals provided sufficiently high levels of TVA to allow for subsequent infection with a modified rabies virus (i.e., SADΔG(envA)). This is a good reminder of the extremely high efficiency of this rabies tracing system and simultaneously reinforces its limitations: although some studies have used this system to examine inputs to ubiquitous cell types such as GABA neurons (*Watabe-Uchida et al., 2012*; *Weissbourd et al., 2014*; *Beier et al., 2015*), there is substantial risk for this adeno-associated virus (AAV) to infect not only Cre-expressing neurons in the injection site but also Cre-expressing neurons that project to that area, which may result in direct (i.e., non-transsynaptic) infections by the rabies virus. Proper controls are always necessary for tracing using this system, especially because a very small TVA infection could allow for subsequent rabies infection.

## Automating whole brain rabies tracing

The acquisition and analysis of whole-brain tracing data has previously been accomplished by physically sectioning the brain at ~100 μm intervals, imaging each (or every third) slice, manually defining regional boundaries in each section, and counting the number of cells in each region. This process is highly time-consuming and requires the dedicated attention of an expert in brain anatomy. To acquire and analyze this type of data with higher throughput, we decided to make use of novel methods in imaging and analysis.

Several methods have recently been developed with the goal of acquiring an image of the whole mouse brain in an automated fashion. In broad terms, most strategies either employ serial sectioning 2-photon microscopy (*Ragan et al., 2012*; *Oh et al., 2014*) or optical clearing (*Chung and Deisseroth, 2013*) followed by light-sheet microscopy (*Tomer et al., 2014*). Serial sectioning 2-photon microscopy has the advantage of using an opaque brain as starting material, meaning that it doesn't require any clearing steps. Light-sheet imaging of cleared tissues, by contrast, allows for much more rapid image acquisition since there is no time spent physically sectioning the tissue and also because scanning is done with a sheet rather than a line. This increased scan speed allows for the rapid imaging of whole mouse brains with sufficient resolution to image every neuron. Because we wanted to characterize the distributions of all labeled neurons across brains, we decided to use light sheet microscopy on cleared brains.

Among methods to clear brains, a key distinction is between protocols that employ an organic solvent to reduce the internal differences in refractive index within a tissue and those that use an aqueous solution to wash away lipids and thereby reduce total scattering. Methods relying on organic solvents (such as iDISCO) denature native fluorescent proteins and require subsequent antibody staining (*Renier et al., 2014*). Because we wanted to visualize GFP-labeled neurons, we decided to use CLARITY so that some native fluorescence (i.e., GFP and tdTomato) would remain intact, although we found that CLARITY causes a loss of mCherry and BFP fluorescence. Furthermore, exposure to organic solvents dramatically alters the size and overall morphology of the brain, while CLARITY leaves it relatively intact. Therefore, although the autofluorescence is dimmer in CLARITY-cleared brains (compared to iDISCO-cleared brains), brain structures are easier to identify. In summary, we found that the combination of CLARITY and light sheet microscopy allows for the rapid acquisition of whole brain rabies-tracing results.

Paired with automated cell detection and alignment between brains, this method allows for a consistent, unbiased, and automated analysis of tracing experiments that would previously have required countless hours of attention from an expert in brain anatomy.

## Conclusions and future directions

Although the neural circuits controlling dopamine function are complex, a comprehensive understanding of the inputs/output structure of different populations of dopamine neurons will likely prove useful for understanding these circuits. In this study, we contributed to this effort in three different ways. First, we developed an automated pipeline for whole brain imaging, using a brain-clearing method (CLARITY), light sheet microscopy, and semi-automated data analysis. This will help to lower the hurdle for future

systematic anatomy studies, and increase their consistency and efficiency. Second, we defined the full sets of monosynaptic inputs to eight subpopulations of dopamine neurons specified by projection targets. Third, we found that dopamine neurons projecting to the tail of the striatum are unique in their monosynaptic inputs, suggesting that they might also be an outlier functionally compared both to dopamine neurons projecting to other parts of the striatum and to those projecting to other brain regions (i.e., cortex, GP, amygdala). This foundational work also opens the door to further investigations. For example, what are the functional differences between the anterior and posterior (tail) regions of the striatum, and what is the role of dopamine in each? What information is calculated by dopamine neurons that project to different parts of the striatum, and how do the patterns of their inputs account for the differences in their signals? How is each dopamine population related to the symptoms of various disorders such as Parkinson's disease? We hope that our work will help guide these types of studies by providing them with a detailed circuit diagram as a starting point.

## Materials and methods

### Animals

86 male adult mice were used. These mice were the result of a cross with C57BL/6J mice and DAT (*Slc6a3*)-Cre mice such that they were heterozygous for Cre recombinase under the control of the dopamine transporter (DAT) (*Backman et al., 2006*). Four male Vglut2 (*Slc17a6*)-Cre (*Vong et al., 2011*) crossed with B6.Cg-*Gt(ROSA)20Sor*^tm9(CAG-tdTomato)Hze/J (Jackson Laboratory, Bar Harbor, ME, United States) were used to help with the visualization of region boundaries. All procedures were in accordance with the National Institutes of Health Guide for the Care and Use of Laboratory Animals and approved by the Harvard Animal Care and Use Committee.

### Viral injections

pAAV-EF1α-FLEX-TVA was made by inserting TVA950 (*Belanger et al., 1995*) truncated after transmembrane domain in pAAV-EF1α-FLEX (*Watabe-Uchida et al., 2012*). pAAV-CA-BFP was made by cloning mTagBFP (Evrogen, Moscow, Russia) (*Subach et al., 2008*) in pAAV-CA, which was made by removing FLEX from pAAV-CA-FLEX (*Watabe-Uchida et al., 2012*). AAVs were produced by the UNC Vector Core Facility (Chapel Hill, NC, United States).

In the first surgery, we injected 250–500 nl of AAV into the projection site (AAV5-FLEX-TVA, $1 \times 10^{12}$ particles/ml and AAV1-BFP, $9 \times 10^{12}$ particles/ml) and either AAV8-FLEX-RG ($2 \times 10^{12}$ particles/ml) (*Watabe-Uchida et al., 2012*) (to label inputs), or no virus (to label only starter neurons) into both the VTA and SNc. After 3 weeks, in the second surgery, 250–500 nl of SADΔG-EGFP(EnvA) ($5 \times 10^7$ plaque-forming units [pfu]/ml) (*Wickersham et al., 2007*) was separately injected into both the VTA and SNc. Brain samples were collected after 1 week. All surgeries were performed under aseptic conditions with animals anesthetized with isoflurane (1–2% at 0.5–1.0 l/min). Analgesia (ketoprofen, 5 mg/kg, I.P.; buprenorphine, 0.1 mg/kg, I.P.) was administered for the 3 days following each surgery. The virus injection sites are as follows: (VTA) Bregma −3.1, Lateral 0.6, Depth 4.3 to 4.0, (SNc) Bregma −3.1, Lateral 1.2, Depth 4.3 to 4.0, (VS) Bregma 1.3, Lateral 0.5, Depth 3.8, (DS) Bregma 1.0, Lateral 1.5, Depth 2.8, (TS) Bregma −1.7, Lateral 3.0, Depth 2.35, (mPFC) Bregma 2.1, Lateral 0.75, Depth 1.5, Angle 20°, (OFC) Bregma 2.45, Lateral 1.35, Depth 1.75, (Amy) Bregma −1.35, Lateral 2.35, Depth 4.0, (lHb) Bregma −1.8, Lateral 1.15, Depth 2.95, Angle 15°, (GP) Bregma −0.45, Lateral 1.85, Depth 3.5.

### Perfusion in hydrogel

To prepare a hydrogel solution, water, paraformaldehyde (PFA), phosphate buffered saline (PBS), Bis, and Acrylamide were mixed on ice, and then VA-044 Initiator (Fisher Scientific NC9952980, Waltham, MA, United States) was added. Then, 50 ml aliquots were made for storage at −20°C. Hydrogel was thawed at −4°C for 3 hr prior to each perfusion. During the perfusions, hydrogel solution was kept on ice. Immediately after the perfusions, brains were transferred to −4°C. The recipe used for hydrogel:

> 40 ml of 40% Acrylamide
> 10 ml of 2% Bis
> 1 gram VA-044 Initiator
> 40 ml of 10× PBS
> 100 ml of 16% PFA
> 210 ml of $H_2O$

## Oxygen/nitrogen exchange and hydrogel solidification

After 2 days of storage in hydrogel at 4°C, oxygen was removed from tube using a food sealer and replaced with nitrogen. Immediately after oxygen/nitrogen exchange, tubes were placed in a water bath at 37°C. After 1.5 hr, the hydrogel solution solidified into a jelly-like texture. At this point, brains were gently removed from the hydrogel using gloved fingers. Brains were placed on a paper (Kimwipe, Kimtech Science, Franklin, MA, United States) and gently rolled over the paper so that excess hydrogel would come off of the brain. Great care was taken to ensure that all of the hydrogel was removed, but that none of the brain was damaged in the process. At this point, brains were put in PBST (0.1% Triton in PBS, pH = 8) overnight at 37°C on a rocker, covered by aluminum foil. After 1 day, brains were transferred to clearing solution. The recipe used for clearing solution:

123.66 g Boric Acid
400 g SDS
9.5 l $H_2O$
60 ml NaOH (10 M)

Water, boric acid, and NaOH were mixed in a 5-gallon plastic container. Then, sodium dodecyl sulfate (SDS) was added 100 g at a time while mixing. Finally, solution was left to mix for several hours on a rocker at 37°C prior to use.

## Tissue clearing

Clearing was based on the original CLARITY protocol (*Chung and Deisseroth, 2013*). Simple and inexpensive parts were used in place of the originally proposed components. A Niagra 120 V (Grey Beard Pumps #316, Mt Holly Springs, PA, United States) pump was used to circulate clearing solution. A Precision Adjustable 60 V/5 A (Korad Technology #KA6005D, Shenzhen, China) power supply was used to provide current at a constant voltage. A 5-gallon plastic container (US Plastic #97,028, Lima, Ohio, United States) was used as a clearing solution reservoir and tubing was run though a second 5-gallon plastic container filled with water to cool the solution flowing through it. Chambers were constructed as previously described (*Chung and Deisseroth, 2013*) using a Nalgene chamber (Nalgene 2118-0002, Rochester, NY, United States) and platinum wire (Sigma-Aldrich 267228, St. Louis, MO, United States). Clearing was done in a room held at 37°C.

Electrophoresis was performed with a constant current of 30 V, with an average temperature of 40°C, over the course of 60 hr. With the voltage held at 30 V, the current fluctuated between 0.5 A and 1 A, and generally stabilized at ~0.75 A. The exact voltage/current relationship varied slightly depending on the chamber. The polarity of the electric field was switched every 12 hr. Clearing solution was replaced after every 120 hr of use (so, after using it twice). Platinum wire was replaced after every 1200 hr of use (so, after using it 20 times). After removing brains from the clearing chamber and placing them in PBST (0.1% Triton, pH = 8) for 24 hr, brains appeared snow white and expanded relative to their normal size.

## Preparation for imaging

Brains were placed in an imaging solution called OptiView (*Isogai et al., in press*) (patent pending, application U.S. Serial No. 62/148,521) with refractive index 1.45 and pH 8 at room temperature on a rocker for 2 days before imaging. The imaging solution used was very similar to the recently described 'RIMS' imaging solution (*Yang et al., 2014*). RIMS solution can be made as follows:

40 grams of Histodenz (Sigma)
30 ml PBS
NaOH to pH 7.5
RI to 1.46

## Light-sheet imaging

Images were acquired with the Zeiss Z.1 Lightsheet microscope (Carl Zeiss, Jena, Germany). Brains were glued to the tip of a 1 ml syringe (without needle) such that the posterior tip of the cerebellum was in contact with the syringe. Brains were then lowered into the imaging chamber. A 488 nm laser was used to excite GFP and a 561 nm laser was used to produce autofluorescence. Images were collected through a 5× objective with PCO-Edge scMOS 16 bit cameras (PCO, Kelheim, Germany) with 1920 × 1920 pixels. Each frame was 2000 × 2000 μm, so each pixel was roughly 1.04 μm. The step

size between images was set to 5.25 µm, so the voxels were not quite isotropic. Brains were imaged horizontally from the dorsal side, and then rotated 180° for imaging from the ventral side. Each view was tiled with 7 × 6 tiles (14,000 × 12,000 µm) and the two views were combined to create a continuous image. Autofluorescence images were subsequently downsized to 700 × 600 × 350 pixels for alignment to the reference space. In these downsized images, voxels have 20 µm spacing in all 3 dimensions.

## Cell detection

Images were segmented into 'cell' and 'non-cell' pixels with eight different segmentation algorithms. Each algorithm was trained with a selection of images from one of eight 'parent regions' including: (1) olfactory bulb, (2) cortex/hippocampus, (3) thalamus, (4) striatum/pallidum, (5) hypothalamus, (6) midbrain, (7) hindbrain, and (8) cerebellum. For each of these parent regions, a random set of 50 images from 10 different brains were used to train each of the segmentation algorithms manually using Ilastik, software that has been recently applied to segment EM images (*Sommer et al., 2011*; *Maco et al., 2014*). Because neuron appearance and background fluorescence appearance differs drastically through the brain, segmentation algorithms only reliably recognized cells within the region they were trained on.

Ilastik was used to calculate six features for each pixel (Gaussian, Laplacian of Gaussian, Gaussian Gradient Magnitude, Difference of Gaussians, Structure Tensor Eigenvalues, Hessian of Gaussian Eigenvalues) with 3 radiuses (0.7 pixels, 1.6 pixels, and 5 pixels). In total, therefore, a vector with 18 values was produced for each pixel. During training, subsets of these vectors (pixels) would be marked as 'cell' or 'non-cell' by a human in each of the training images and then the rest of the vectors' identities were inferred using a random forest with 200 trees and four randomly chosen features per tree. Random sets of images from brains that were not part of the training set were used to manually verify the accuracy of segmentation.

After training, all eight segmentation algorithms were applied to all of the images. This resulted in a binary version of the original data from the microscope with all pixels either classified as '1 (cell)' or '0 (non-cell)'. Because each cell was present in multiple Z positions, all sets of neighboring pixels that were labeled as '1 (cell)' pixels were collapsed in 3D using MATLAB to find the centroid of each detected cell. This afforded an extra opportunity to reduce noise by only counting 'cells' that were within a reasonable range of diameters (2.5 µm–50 µm), present in consecutive images, and with a reasonable shape (i.e., circularity of >0.1). Finally, the centroids' positions were transformed into the reference space based on the result of autofluorescence alignment (see below). At this point, the centroids were masked using the outline of the appropriate parent region and combined. For example, only the results of the cortex segmentation algorithm within the cortex were retained.

## Brain alignment

The autofluorescence signals derived from 25 brains were averaged to produce a single 'reference space' to serve as a template for subsequent brains. Subsequent brains were routinely aligned to this reference space using Elastix (*Klein et al., 2010*). We performed affine alignment followed by B-spline alignment based on mutual information, as previously proposed for human MRI image registration (*Metz et al., 2011*). The resulting transformation (from individual brain to reference space) was applied to the centroids of detected cells in order to plot all cells in the same reference space.

## Atlas construction

First, the 100 coronal slices of the Franklin and Paxinos atlas (*Franklin and Paxinos, 2008*) were manually aligned to the nearest matching optical slices of the reference brain (which has 700 coronal optical sections total). Then, boundaries of major regions were manually drawn onto each of these 100 slices. Finally, each region was manually smoothed in each dimension to produce continuous regions in 3D.

## Immunohistochemistry

Brains were sectioned using a vibrotome with 1 mm spacing. After CLARITY, brains were extremely difficult to section owing to their unusual structural properties (soft, yet resistant to cutting). Slices were stained as previously described (*Chung and Deisseroth, 2013*). Briefly, primary antibody was applied for 2 days (then washed off for 1 day) and then secondary antibody (Molecular Probes, Eugene, OR, United States) was applied at 1:200 for 1 day (then washed off for 1 day). Every step (including washes) was performed at 37°C on a rocker with PBST (0.1% Triton, pH = 8) containing sodium borate (1 M).

## Confocal imaging

After CLARITY, BFP signal was lost completely. A rabbit anti-tRFP antibody (Evrogen #234, 1:100) was used to detect BFP signal and a chicken anti-TH antibody (Abcam #76442, 1:100, Camgridge, United Kingdom) was used to detect TH. Brains were imaged to determine injection site and percentage of TH-positive starter cells using an inverted confocal microscope (Zeiss LSM 700). Stained 1 mm sections were placed in a glass-bottom well and then covered with a thin layer of imaging solution (as described above) to prevent them from drying during image acquisition. Images were acquired and stitched using Zeiss Blue.

## Data analysis

Further data analysis was performed using custom software written in MATLAB (Mathworks, Natick, MA, United States). For statistical comparisons of the mean, we used 1-way analysis of variance (ANOVA) and the two-sample Student's *t*-test, unless otherwise noted. The significance level was corrected for multiple comparisons using a Holm-Sidak method. All error bars in the figures are the standard error of the mean (s.e.m.).

To quantify the percentage of inputs, we excluded areas that contained dopamine cell bodies that could potentially have been starter cells in and around VTA, SNc, SNr and RRF, to avoid potential contamination from dopamine neurons in the counts of inputs. We defined the primary infection sites by adding the sites of all the dopamine cell bodies in experiments without RG (*Figure 1A*, *Figure 3—figure supplement 1*). We did not exclude A10rv in the supramammillary area because DAT positive neurons in this area projected only to Amy and data did not change with or without this area.

To quantify the similarity in input patterns, we calculated Pearson's correlation coefficients between the percent of input neurons across anatomical areas. To compare spatial distributions of input neurons in the forebrain, the mean of the coordinates of all input neurons on a given coronal section was obtained. Hierarchical clustering was done using the correlation coefficients as a distance metric and using the average linkage function. Using other linkage functions produced similar results.

To define 'ventral patches', we first pooled all neurons from all animals and plotted them in a reference space with 20 μm × 20 μm × 20 μm voxels. A 3D Gaussian with kernel size of 60 μm × 60 μm × 60 μm was then used to estimate the density of cells at each voxel. We defined ventral patches by finding the local maxima of the Gaussian-filtered 3D data and expanding stepwise pixel-by-pixel until either 1/3 of the maximum intensity or another patch boundary was reached.

## Acknowledgements

We thank Ju Tian, Clara Starkweather, Catherin Dulac and other members of the Uchida lab and Dulac lab for helpful discussions. We thank Lingjin Zheng for pilot experiments. We are grateful to Drs Linh Vong and Bradford Lowell for Vglut2-Cre mice. We also thank the staff of the Harvard FAS Research Computing Group as well as the staff of the Harvard Center for Biological Imaging for technical support.

## Additional information

### Competing interests

JFB: Yoh Isogai and Joseph Bergan have filed a patent application on OptiView. YI: Yoh Isogai and Joseph Bergan have filed a patent application on OptiView. NU: Reviewing editor, *eLife*. The other authors declare that no competing interests exist.

### Funding

No external funding was received for this work.

### Author contributions

WM, Conception and design, Acquisition of data, Analysis and interpretation of data, Drafting or revising the article; JFB, YI, KUV, PO, Techniques for data acquisition and analysis; SKO, Acquisition of data, Analysis and interpretation of data; NU, MW-U, Conception and design, Analysis and interpretation of data, Drafting or revising the article

### Ethics

Animal experimentation: This study was performed in strict accordance with the recommendations in the Guide for the Care and Use of Laboratory Animals of the National Institutes of Health. All of the

animals were handled according to approved Harvard animal care and use committee (IACUC) protocols (#26-03) of Harvard University. All surgery was performed under isofluorane anesthesia, and every effort was made to minimize suffering.

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
