## [Decision Letter]

Thank you for submitting your work entitled “Dopamine neurons projecting to the posterior striatum form an anatomically distinct subclass” for peer review at *eLife*. Your submission has been favorably evaluated by Eve Marder (Senior Editor), Sacha Nelson (Reviewing Editor), and three reviewers.

The reviewers have discussed the reviews with one another and the Reviewing Editor has drafted this decision to help you prepare a revised submission.

Using sophisticated viral tracing techniques and the new anatomical technique, CLARITY, the authors present an unprecedentedly detailed look at the correlation between input and output projections of dopamine neurons. The main findings are that inputs tend not to be reciprocal, but rather most subpopulations receive broad, overlapping inputs. Interestingly, dorsal posterior projecting dopamine neurons seem to be the only outlier receiving a smaller percentage of inputs from ventral striatum, and a high percentage of inputs from subthalamic nucleus, and zona incerta. These findings provide important information for future studies focused on understanding computations and processing of reward signals that occur in different subpopulations of dopamine neurons.

Essential revisions:

All of the reviewers agreed that further discussion and clarification could improve the paper, but that this should not require further experiments.

1) At least two of the reviewers felt that the issue of whether or not DA neurons project to multiple targets was an important one for the field that your manuscript could potentially address, at least in the Discussion.

2) The reviewers also felt that more precise definition of the tail of the striatum would be helpful.

3) The reviewers noted a similar manuscript that has recently been published in Cell, but it is *eLife* policy not to hold papers published during the review process as discounting the novelty of the study; furthermore the reviewers felt that the findings may be complementary and that this was good for the field. You should however cite this paper in the final version of the manuscript.

4) Finally, at least one of the reviewers and the Reviewing Editor felt the term “inputome” was cumbersome and should be removed.

Although the full reviews of each of the three reviewers are included below for your information, there are some points that you may choose not to address in the revision.

Reviewer #1:

This study by Menegas et al. is a tour de force in data acquisition and analysis that will become a highly cited, a landmark paper describing the anatomical organization of the dopamine system. Using sophisticated viral tracing techniques and the exciting new technique, CLARITY, the authors present an unprecedentedly detailed look at the correlation between input and output projections of dopamine neurons. It is a daunting task to summarize all their findings in a concise and coherent manuscript, and overall the authors do well, but a few changes could improve presentation of the data.

1) One of the main results the authors discuss in detail is the finding that dopamine neurons projecting to the “tail” of the striatum receive unique patterns of inputs compared to other populations of DA neurons. It would be nice if the boundaries of this “tail” could be better described and displayed in a figure. It is described as posterior, suggesting there is an A-P gradient within the striatum, which would be best shown in the sagittal plane. But no sagittal images are shown.

2) The authors cite several studies presenting conflicting data about whether DA neurons project to one target brain area or several. Using CLARITY, shouldn't the authors be able to answer this long-standing question? If axons from a “DS” dopamine neuron are seen in a different brain area, this demonstrates that DA neurons project to multiple targets.

3) Statements are made in the Introduction about differences in innervation to SNc vs VTA dopamine neurons, but explicit comparisons of these differences are lost in the presentation of data. Indeed, summary Figure 9 lumps VTA and SNc together. This should be clarified.

4) Data from Figure 3 would seem to argue there are no clear spatially-defined zones in the SNc. Is it possible to create a collapsed and/or 3D image of distribution of distinct DA neuron populations within SNc to summarize the results of this figure? Also, where is the SNr in this figure? How were the borders of DA neurons in the SNc distinguished from dopamine neurons in the SNr?

5) How long after rabies injections were animals sacked? I can see that constructs with RG and TVA were injected 3 weeks before rabies, but it is not clear how long animals survived after rabies injections.

6) The term “inputome” in the Introduction is odd and not necessary.

7) In Figure 7—figure supplement 1, no legend is listed for the different colored patches. Presumably they are the same as Figure 7, but this should be specified.

Statistics seem to be quite rigorous

Reviewer #2:

This manuscript examines the afferent inputs to dopamine neurons that project to selective areas of the CNS. The specific projection areas were well chosen, the array of afferent inputs is comprehensive and for the most part the results were well presented. There may be technical aspects of this study that this reviewer would not understand however from my point of view the results are convincing, unexpected and will be of interest to a wide range of neuroscientists.

The most interesting and exciting observation in the study is the distinct distribution of inputs from the accumbens. The five distinct patches are amazing. Even more interesting is that dopamine cells that receive input from one or another of these patches project to distinct areas.

1) Parts of the results need more explanation:

a) At the very end of the subsection “Distribution of dopamine neurons projecting to different projection targets”, the explanation of why spread of infection between dopamine neurons was not a problem is difficult to understand. Is there potential for infection through dendro-dendritic contacts? It would be helpful if there was more explanation of this point.

b) Some of the figure legends were too brief. For example – what are the black bars on the plots at the bottom of Figure 6? The colored ones make sense. It was difficult to link the various parts of supplemental figure 3. More explanation would be helpful.

2) The Introduction is over-written. The primary point of the manuscript is addressed in the last two paragraphs. It seems that much of the first two paragraphs could be very much condensed and the third paragraph could be eliminated completely.

3) The Discussion is speculative. It seems that all discussion of reward prediction error is very much out of place in a strictly anatomical study. Likewise, the link of this work to Parkinson's disease does not contribute very much. The anatomical aspects of this manuscript are novel and super interesting. The Discussion should be aimed at the strengths of the study.

Reviewer #3:

This manuscript by Menegas et al. presents a neuroanatomical study of monosynaptic inputs into midbrain dopamine neuron subpopulations. In their past work (100), the authors made a fundamental contribution to the field by using retrograde viral tracing to produce a whole-brain map of synaptic inputs into dopamine neurons. They showed that VTA and SNc dopamine neurons receive an incredibly diverse array of inputs from a wide range of brain nuclei. In the present study, they focus on the much more difficult question of whether the patterns of synaptic innervation vary between subpopulations dopamine neurons categorized according to their targeted regions. To address this question, the authors use advanced viral tracing and brain clearing techniques. They developed a pipeline which included semiautomated acquisition and analysis of data from cleared brains acquired using light-sheet microscopy.

This is an excellent study that presents a number of novel findings which I think will prove to be fundamental in the field. The main finding here is that inputs to dopamine neurons do not arrive preferentially from their targeted regions (reciprocal), but rather most subpopulations receive broad, overlapping inputs. Interestingly, dorsal posterior projecting dopamine neurons seem to be the only outlier receiving a smaller percentage of inputs from ventral striatum, and a high percentage of inputs from subthalamic nucleus, and zona incerta. These findings provide important information for future studies focused on understanding computations and processing of reward signals that occur in dopamine neurons of different subpopulations.

I have only minor comments:

1) Of the positively labeled neurons within the SNr (Figure 3), what portion of these are retrogradely labeled versus directly infected by the rabies virus? The authors may want to discuss more why this region was not quantified.

2) It was also surprising that the largest percentage of inputs to mostly all subpopulations arrive from lateral hypothalamus (Figure 4). This seems different from the findings of [100] that showed striatum as the most dominant input. Is this difference methodological or in the way they were quantified? A brief discussion of the seeming discrepancy between the differences in quantification may help the reader.

3) The centroid analysis of Figure 6 is interesting but the presentation leaves the reader in search of a broader meaning/framework/context. Does this finding provide insight in the development of the system or something else? Whatever the importance, a brief discussion of this would be useful to guide the reader.

---

## [Author Response]

*1) At least two of the reviewers felt that the issue of whether or not DA neurons project to multiple targets was an important one for the field that your manuscript could potentially address, at least in the Discussion*.

We understand that this is an important issue. Unfortunately, our method was not well suited to answer this question. We decided to use standard eGFP rather than a membrane-targeted GFP because we wanted to make automated segmentation as straightforward as possible (i.e. minimize the signal from axons). Further, as performed in our lab, CLARITY does not preserve weak signals, including signals from thin axons. For these reasons, it was difficult to follow axon signals precisely in our samples.

In the past, many studies tried to address this question, and there are controversies regarding the level of collateralization of dopamine neurons. We originally discussed this issue briefly in the Results section. Now, we have expanded our thoughts and moved it to the Discussion, subsection “Technical considerations and relation with recent studies”.

*2) The reviewers also felt that more precise definition of the tail of the striatum would be helpful*.

Thank you for pointing this out. We used the term “tail” to describe the most posterior part of the striatum. We chose the coronal plane for this injection at Bregma -1.7. Because all BFP signal was lost during the CLARITY-based clearing process, the precise injection sites were examined in coronal sections. As a reviewer suggested, we have added an illustration of compressed horizontal and sagittal planes in Figure 3—figure supplement 5.

*3) The reviewers noted a similar manuscript that has recently been published in Cell, but it is* eLife *policy not to hold papers published during the review process as discounting the novelty of the study; furthermore the reviewers felt that the findings may be complementary and that this was good for the field. You should however cite this paper in the final version of the manuscript*.

We appreciate that the reviewers felt that our findings are important for the field. We agree that the studies are complementary. These groups also found that dopamine neurons with different projection targets have similar inputs, although none of them examined dopamine neurons projecting to the tail of the striatum. We added a reference to both of these papers to our Discussion, in the subsection “Technical considerations and relation with recent studies”.

4) Finally, at least one of the reviewers and the Reviewing Editor felt the term “inputome” was cumbersome and should be removed.

We removed this term entirely from the manuscript and replaced it with more descriptive phrases that explain what we were trying to convey (i.e. “set of monosynaptic inputs”).

Reviewer #1:

*1) One of the main results the authors discuss in detail is the finding that dopamine neurons projecting to the “tail” of the striatum receive unique patterns of inputs compared to other populations of DA neurons. It would be nice if the boundaries of this “tail” could be better described and displayed in a figure. It is described as posterior, suggesting there is an A-P gradient within the striatum which would be best shown in the sagittal plane. But no sagittal images are shown*.

Please see our response to the second point of the essential revisions.

*2) The authors cite several studies presenting conflicting data about whether DA neurons project to one target brain area or several. Using CLARITY, shouldn't the authors be able to answer this long-standing question? If axons from a “DS” dopamine neuron are seen in a different brain area, this demonstrates that DA neurons project to multiple targets*.

Please see the response to essential revision 1.

*3) Statements are made in the Introduction about differences in innervation to SNc vs VTA dopamine neurons, but explicit comparisons of these differences are lost in the presentation of data. Indeed, summary*
Figure 9
*lumps VTA and SNc together. This should be clarified*.

In our previous study, we showed that dopamine neurons in VTA receive different inputs from dopamine neurons in SNc. However, in the present study, we injected AAV-Flex-RG and rabies virus into both the VTA and SNc in every animal, meaning that we distinguished the neurons based purely on their projection site rather than based on their physical location. We found that each population of neurons (with the same projection site) included neurons in both VTA and SNc. We clarified this point in the Results and Discussion sections.

*4) Data from*
Figure 3
*would seem to argue there are no clear spatially-defined zones in the SNc. Is it possible to create a collapsed and/or 3D image of distribution of distinct DA neuron populations within SNc to summarize the results of this figure? Also, where is the SNr in this figure? How were the borders of DA neurons in the SNc distinguished from dopamine neurons in the SNr?*

The boundaries between VTA, SNc and SNr may not be clear in Figure 3—figure supplement 1 because we have projected (collapsed) many coronal planes (400 µm) in each panel to summarize the data with a small number of images. SNr is located below and to the right of the red-most area (dopamine neurons-concentrated area) in each plane. We added “SNr” in the figure for clarification. We did not distinguish dopamine neurons in SNc and SNr, but instead excluded both areas from statistical analysis to avoid contamination of starter dopamine neurons.

We could not define dorsal and ventral tiers in SNc clearly, either because of the difference between rats and mice or because we lacked sufficient resolution or proper markers. However, we observed a difference in the distribution of dopamine subpopulations along the medial-lateral axis. We added Figure 3—figure supplement 2 showing collapsed (and normalized) images of the entire area for comparison.

5) How long after rabies injections were animals sacked? I can see that constructs with RG and TVA were injected 3 weeks before rabies, but it is not clear how long animals survived after rabies injections.

We waited for one week after rabies injection. This is the same amount of time that we waited in our previous studies. We added this information in the Results and Methods sections.

*6) The term “inputome” in the Introduction is odd and not necessary*.

Please see our response to essential revision 4.

*7) In*
Figure 7—figure supplement 1, *no legend is listed for the different colored patches. Presumably they are the same as*
Figure 7*, but this should be specified.*

We added more explanation to the legend of Figure 7—figure supplement 1. It was indeed the same color-coding as in Figure 7.

Reviewer #2:

*1) Parts of the results need more explanation*:

a) At the very end of the section titled Distribution of dopamine neurons projecting to different projection targets. The explanation of why spread of infection between dopamine neurons was not a problem is difficult to understand. Is there potential for infection through dendro-dendritic contacts? It would be helpful if there was more explanation of this point.

We estimate that transsynaptic spread between dopamine neurons did not affect our results for three reasons: (1) previous literature suggested that dopamine neurons are not heavily connected each other but connected with other cell types, (2) the distributions of labeled neurons with or without transsynaptic labeling in the VTA/SNc are similar, and (3) the published works that overcame this issue resulted in similar results. Although the level of dendro-dendritic contacts and its effects on rabies transmission are not clear, (2) and (3) support validity of our data. We discussed this further in the Discussion section.

*b) Some of the figure legends were too brief. For example - what are the black bars on the plots at the bottom of*
Figure 6*? The colored ones make sense*. *It was difficult to link the various parts of supplemental figure 3. More explanation would be helpful*

We expanded the descriptions in the figure legends, ensuring (in particular) that all scale bars are labeled. Thank you for catching this mistake. We also expanded the figure legends for the supplements of Figure 3.

*2) The Introduction is over-written. The primary point of the manuscript is addressed in the last two paragraphs. It seems that much of the first two paragraphs could be very much condensed and the third paragraph could be eliminated completely*.

We deleted some sentences in the Introduction.

*3) The Discussion is speculative. It seems that all discussion of reward prediction error is very much out of place in a strictly anatomical study. Likewise, the link of this work to Parkinson's disease does not contribute very much. The anatomical aspects of this manuscript are novel and super interesting. The Discussion should be aimed at the strengths of the study*.

We tried to make it clear that these parts of the Discussion are speculative, made them more concise, and added discussion specifically about anatomy studies.

Reviewer #3:

*1) Of the positively labeled neurons within the SNr (*Figure 3*), what portion of these are retrogradely labeled versus directly infected by the rabies virus? The authors may want to discuss more why this region was not quantified*.

We found many fewer neurons in the SNr when we labeled dopamine neurons only, based on their projection site (i.e. in cases where we did not allow for spread to inputs). Therefore, we believe that labeled neurons in SNr are primarily inputs. However, we did not quantify the number of input neurons in any of the dopamine-containing nuclei (VTA, SNc, SNr, RRF) so that we knew for a fact that we were not counting starter neurons as input neurons. We added more detail in the Methods section to make this point clear.

*2) It was also surprising that the largest percentage of inputs to mostly all subpopulations arrive from lateral hypothalamus (*Figure 4*). This seems different from the findings of*
[100]
*that showed striatum as the most dominant input. Is this difference methodological or in the way they were quantified? A brief discussion of the seeming discrepancy between the differences in quantification may help the reader*.

In our previous study, we injected rabies virus either into the VTA or SNc, whereas in our current study we injected rabies virus both into the VTA and SNc. Therefore, comparing them directly is difficult. We more clearly stated this point in the Results, Methods, and Discussion sections.

We previously observed that LH, as well as DS and VS, was one of the biggest sources of inputs to VTA dopamine neurons whereas DS was by far the largest source of input to SNc dopamine neurons. This time, we identified LH and DS as some of the biggest sources of inputs to most dopamine subpopulations. It is possible that the difference of inputs in the previous study may arise from location (SNc) rather than specific projection targets. Because it is important to provide accurate information, we added this discussion.

*3) The centroid analysis of*
Figure 6
*is interesting but the presentation leaves the reader in search of a broader meaning/framework/context. Does this finding provide insight in the development of the system or something else? Whatever the importance, a brief discussion of this would be useful to guide the reader*.

The fact that monosynaptic inputs are topographically organized is potentially important but we do not know the meaning of this organization yet. We made this point clearer in the Results section.
